# SimpliHuMoN: Simplifying Human Motion Prediction

## Abstract

Human motion prediction combines the tasks of trajectory forecasting, human pose prediction, and possibly also multi-person modeling. For each of the three tasks, specialized, sophisticated models have been developed due to the complexity and uncertainty of human motion. While compelling for each task, combining these models for holistic human motion prediction is non-trivial. Conversely, holistic human motion prediction methods, which have been introduced recently, have struggled to compete on established benchmarks for individual tasks. To address this dichotomy, we study a simple yet effective model for human motion prediction based on a transformer architecture. The model employs a stack of self-attention modules to effectively capture both spatial dependencies within a pose and temporal relationships across a motion sequence. This simple, streamlined, end-to-end model is sufficiently versatile to handle pose-only, trajectory-only, and combined prediction tasks without task-specific modifications. We demonstrate that our approach achieves state-of-the-art results across all tasks through extensive experiments on a wide range of benchmark datasets, including Human3.6M, AMASS, ETH-UCY, and 3DPW. Our results challenge the prevailing notion that architectural complexity is a prerequisite for achieving accuracy and generality in human motion prediction. Code will be released.

## 1 Introduction

Human motion prediction, the task of forecasting future 3D human motion from a sequence of past observations, is a critical challenge with wide-ranging applications in autonomous driving (Zheng et al., 2022; Paden et al., 2016), robotics (Zou, 2024; Salzmann et al., 2023), virtual reality (Clark et al., 2020; Fu et al., 2020; Ro et al., 2019), and sports analytics (Li et al., 2021). Because human motion is inherently multi-dimensional, non-linear, and highly uncertain, the literature has largely tackled prediction of human motion by addressing distinct tasks individually: trajectory prediction (Gu et al., 2022; Bae et al., 2022; Shi et al., 2023; Bae et al., 2024; Yao et al., 2024; Fang et al., 2025), pose prediction (Dang et al., 2022; Barquero et al., 2023; Sun & Chowdhary, 2024; Hosseininejad et al., 2025; Curreli et al., 2025; Xu et al., 2024), and multi-person motion prediction (Jeong et al., 2024; Zheng et al., 2025).

While making individual tasks easier to address, this differentiation also opens up a gap: tasks like pose and trajectory forecasting are fundamentally interrelated and governed by the same underlying dynamics (Zheng et al., 2025), yet they are modeled separately using task-specific architectures. This has led to the development of complex, specialized models that excel at one task but struggle to generalize, limiting their applicability and introducing unnecessary complexity. Notable exceptions that jointly model these different tasks, particularly in the context of multi-person motion, are Jeong et al. (2024) and Zheng et al. (2025). However, the results of these holistic models are suboptimal on established benchmarks for individual sub-tasks. Consequently, models that predict jointly tend to create their own benchmarks or evaluation protocols, making it difficult to assess their effectiveness against specialized methods directly. Their performance limitations on pose and trajectory prediction show the need for a solution that not only addresses human motion prediction holistically but also excels on established, task-specific benchmarks.

To achieve this, we present a general and, in hindsight, very simple approach to 3D human motion prediction. Our model is built upon a stack of self-attention modules to effectively capture

both the spatial dependencies within a single pose and the temporal relationships across the entire motion sequence. This design allows us to model a variety of complex motion dynamics while maintaining a streamlined and efficient framework. Unlike more complicated, multi-stage models, our method employs a unified, end-to-end training process, which improves training stability and overall performance. Our findings demonstrate that a well-designed, attention-based model can achieve benchmark performance across all tasks, challenging the notion that architectural complexity is a prerequisite for accuracy and generality in this field.

We validate our approach through extensive experiments on a wide range of public datasets, including Human3.6M (Ionescu et al., 2013) and AMASS (Mahmood et al., 2019) for pose prediction, ETH-UCY (Lerner et al., 2007; Pellegrini et al., 2009) and SDD (Robicquet et al., 2016) for trajectory prediction, as well as MOCAP-UMPM (CMU Graphics Lab, 2003; van der Aa et al., 2011) and 3DPW (von Marcard et al., 2018) for combined pose and trajectory tasks. Our results show that our model outperforms or matches current best methods across various metrics while being computationally efficient.

The key contributions of this paper are summarized as follows:

- We introduce SimpliHuMoN, a simple, unified transformer architecture that can outperform results of complex, specialized human motion prediction model~~unified Transformer framework that challenges the prevailing trend of architectural complexity in human motion prediction~~.
- We establish state-of-the-art performance across pose, trajectory, and holistic prediction tasks ~~, showing that a single, simple architecture can outperform highly specialized models~~.

## 2 SIMPLIHUMON

We propose a simple yet effective 3D human motion prediction model based on a transformer decoder architecture. The model is designed to be as simple as possible, learning a mapping from a person's past movements to their future movements while accommodating various input and output configurations.

The input $X_{\text{past}}$ consists of two components, each over a historical time horizon of $H$ timesteps. On the one hand, the trajectory $T_{\text{past}} \in \mathbb{R}^{H \times 3}$ represents the path of a root joint (*e.g.*, the hip). On the other hand, the relative body pose $P_{\text{past}} \in \mathbb{R}^{H \times M \times 3}$ represents the state of $M$ joints relative to the root joint. Our framework can operate on either of these inputs individually or on both combined: for trajectory prediction, the model only operates on $T_{\text{past}}$; for pose prediction, the model only operates on $P_{\text{past}}$; and for joint pose and trajectory prediction, the model operates on both.

The model aims to predict the corresponding future state $X_{\text{fut}}$, over a prediction horizon of $F$ timesteps. To capture the uncertainty of motion, following prior work (Jeong et al., 2024), the model generates $K$ distinct proposal states, *i.e.*, $X_{\text{fut}} = (X_{\text{fut}}^1, ..., X_{\text{fut}}^K)$. Each proposal $X_{\text{fut}}^k$, $k \in \{1, ..., K\}$, consists of a complete predicted future state. The composition of $X_{\text{fut}}^k$ mirrors that of the input; it can include a future root trajectory $T_{\text{fut}} \in \mathbb{R}^{F \times 3}$, a future relative body pose $P_{\text{fut}} \in \mathbb{R}^{F \times M \times 3}$, or both, depending on what was provided as input.

**Overview of our method.** As illustrated in Fig. 1, our model begins by independently processing the historical observations $X_{\text{past}}$ and a set of learnable query tokens $\mathcal{Q}_{\text{in}} = (\mathcal{Q}_{\text{in}}^1, ..., \mathcal{Q}_{\text{in}}^F) \in \mathbb{R}^{F \times 3}$ into a context tensor $\mathcal{C}$ and a query tensor $\mathcal{Q}$ respectively (Sec. 2.1). A self-attention-based transformer then processes the tensors (Sec. 2.2). Finally, a multi-modal prediction head regresses the decoder's output $Z$ into $K$ distinct trajectories and pose hypotheses to give the final output, $X_{\text{fut}}$ (Sec. 2.3). We describe the training procedure and model configurations in Sec. 2.4.

### 2.1 INPUT PROCESSING AND EMBEDDING MODULE

This module prepares the raw input data for the transformer decoder by normalizing it and mapping it into a shared high-dimensional latent space of dimension $d_{\text{model}}$. The process creates two main tensors: a context tensor, $\mathcal{C}$, from historical observations and a query tensor, $\mathcal{Q}$, from a set of learnable parameters.

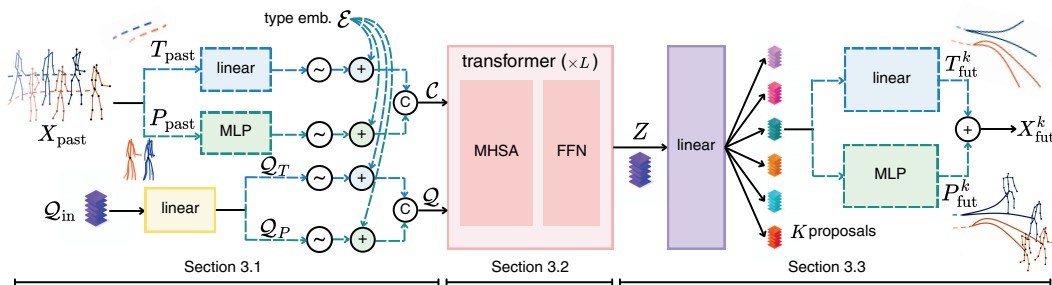

Figure 1: An overview of our architecture. Past observations of 3D poses ($P_\text{past}$) and trajectories ($T_\text{past}$) are jointly processed by an encoder. Learnable input queries ($\mathcal{Q}_\text{in}$), representing potential future states, interact with the encoded past motion within a decoder to produce $K$ distinct future motion proposals ($X_\text{fut}^k$) for all agents over a specified horizon.

### 2.1.1 PAST CONTEXT ENCODING

To compute the context tensor $\mathcal{C}$, the historical input sequence is processed using one or both of two parallel streams, depending on the task: one for trajectory $T_\text{past}$ and one for relative body pose $P_\text{past}$.

**Root Trajectory Processing.** The 3D coordinates of the root joint are extracted from the input sequence. To normalize the motion, the root's position at the final input frame is subtracted from all historical root positions. This normalized trajectory is then projected into the $d_\text{model}$-dimensional embedding space by a linear layer.

**Relative Pose Processing.** The pose is represented relative to the root (hip) joint for each timestep. If a dataset provides absolute coordinates, we normalize the pose by subtracting the root joint's position from all other body joint positions. This relative pose vector is then processed by a two-layer MLP (with a GELU activation function), which outputs an embedding of dimension $d_\text{model}$.

After their initial embedding, both streams are enhanced. First, a sinusoidal positional encoding is added to each sequence to encode the specific position of each of the $H$ timesteps along the time axis. Then, a learnable type embedding $\mathcal{E}$ is added to each token. The type embedding encodes whether a given token represents part of the root trajectory or the body pose. Finally, the processed sequences are concatenated (if both are present) along the sequence dimension to form the final context tensor, $\mathcal{C}$. The shape of $\mathcal{C}$ is therefore $\mathbb{R}^{2H \times d_\text{model}}$ for combined inputs, and $\mathbb{R}^{H \times d_\text{model}}$ when only a single input modality is provided.

### 2.1.2 FUTURE QUERY GENERATION

The queries used to prompt the decoder are learnable tensors $\mathcal{Q}_\text{in} \in \mathbb{R}^{F \times 3}$. Similar to the object queries in DETR (Carion et al., 2020) or learnable soft prompts in language modeling (Lester et al., 2021), these are input-independent parameters optimized during training. They serve as initial "slots" for the $F$ future timesteps, providing the decoder with a temporal structure to fill based on the context. ~~These learnable prompts guide the decoder in its computation.~~ These tokens are first projected into the $d_\text{model}$ space by a linear layer. The resulting sequence is then explicitly split into trajectory $\mathcal{Q}_T \in \mathbb{R}^{F \times d_\text{model}}$ and pose $\mathcal{Q}_P \in \mathbb{R}^{F \times d_\text{model}}$ queries if both modalities are required. Similar to the past context encoding, these query sequences are enriched with positional encodings and their corresponding type embeddings. The two query sequences are then concatenated (if both are present) to create the final query tensor, $\mathcal{Q}$ ($\in \mathbb{R}^{2F \times d_\text{model}}$ for combined inputs, $\mathbb{R}^{F \times d_\text{model}}$ for single), ensuring that it perfectly mirrors the composition and format of $\mathcal{C}$.

This explicit separation of queries into trajectory and pose streams enables the model's flexibility. The architecture learns a strong association between each query type and its corresponding output modality, reinforced by the type embeddings. This allows the same model to handle different tasks without any architectural modifications.

## 2.2 Transformer Decoder

~~The major computations in our model are performed by a decoder-only transformer with $L$ identical layers, utilizing a pre-LayerNorm configuration. Each layer operates on the concatenation of two input tensors: a context tensor, $\mathcal{C}$, derived from historical observations, and a query tensor, $\mathcal{Q}$, derived from learnable latent variables.~~

The core of SimpliHuMoN is a decoder-only transformer that processes historical context and future queries as a single, continuous sequence. Unlike standard encoder-decoder architectures that separate inputs into distinct processing streams connected only by cross-attention, we concatenate the context $\mathcal{C}$ and query $\mathcal{Q}$ tensors along the temporal dimension to form a unified input sequence $[\mathcal{C}; \mathcal{Q}] \in \mathbb{R}^{(H+F) \times d_{\text{model}}}$ for self-attention. ~~A key distinction from standard encoder-decoder or cross-attention-based models is our use of a unified attention mechanism. Within each layer, we perform a single multi-head self-attention operation over the sequence $[\mathcal{C}; \mathcal{Q}]$ concatenated over the time dimension.~~ This design allows every token in the context and query sequences to directly attend to all other tokens, providing a global exchange of information in a single step. For enhanced training stability, we employ pre-LayerNorm with Root Mean Square Layer Normalization (RMSNorm) for training stability and a standard Feed-Forward Network (FFN) with GELU activation. ~~apply Root Mean Square Layer Normalization (RMSNorm) to the query and key projections within each attention head before the dot-product operation. The standard feed-forward network (FFN) sub-layer uses a GELU activation.~~

After passing through the stack of $L$ decoder layers, the model produces an output tensor, $Z$, with the exact dimensions as the input query $\mathcal{Q}$. Having attended to the full context, these output query tokens now serve as rich, context-aware representations ready to be mapped into future predictions by the output heads.

This unified architecture is task-agnostic: whether the input $\mathcal{C}$ contains only trajectory, only pose, or both, the self-attention mechanism naturally adapts to model the available dependencies. ~~The decoder's ability to handle different prediction tasks is a direct consequence of this unified attention design. The architecture is agnostic to the composition of the context $\mathcal{C}$. For combined prediction, the trajectory and pose queries can attend to their corresponding context streams. If the task is trajectory-only, $\mathcal{C}$ will only contain trajectory information, and the query tokens $\mathcal{Q}$ will attend to this relevant context. This allows the model to implicitly specialize its query representations based on the available input, providing a flexible foundation for all task variations.~~

## 2.3 Multi-Modal Prediction Heads

To account for the stochastic nature of the prediction task, the prediction head decodes the final latent representation from the decoder into $K$ distinct future hypotheses. The mechanism is a single linear projection from the decoder's output tensor $Z$ (shape $[F, d_{\text{model}}]$) to an output tensor of shape $[F, K \times C]$, where $C$ is the output dimension (*e.g.* 3 for trajectory, $M \times 3$ for pose). This is then reshaped to $[F, K, C]$, creating $K$ parallel branches. ~~The latent tensor first passes through a linear projection to create $K$ parallel branches.~~ Two dedicated output heads then process each branch, if both are being modeled, to regress the future root trajectory ($T_{\text{fut}}^k$) and body pose ($P_{\text{fut}}^k$), respectively, ensuring each of the $K$ proposals is a complete and comparable hypothesis. Architecturally, these heads mirror the input processing module: a linear layer regresses the trajectory and a two-layer MLP regresses the pose, effectively inverting the initial embedding process.

## 2.4 Implementation Details

The model is trained end-to-end using a "winner-takes-all" loss, where gradients are backpropagated only through the single hypothesis $k$ that minimizes the Euclidean distance to the ground truth future. Formally, the training loss $\mathcal{L}$ for a given ground truth $X_{\text{fut}}^{\text{gt}}$ is computed via

$$\mathcal{L}(X_{\text{past}}, X_{\text{fut}}^{\text{gt}}) = \min_{k \in \{1, \ldots, K\}} \|X_{\text{fut}}^{\text{gt}} - X_{\text{fut}}^k(X_{\text{past}})\|_2, \tag{1}$$

where $X_{\text{fut}}^k(X_{\text{past}})$ is the $k^{\text{th}}$ prediction hypothesis computed from $X_{\text{past}}$ via the model. This formulation ensures that gradients are only computed for the best prediction, encouraging the model's $K$ output modes to specialize and cover diverse, plausible futures.

We report results for two configurations: a "wide" model ($L = 6, d_{\text{model}} = 192$) and a "deep" model ($L = 16, d_{\text{model}} = 48$). In all experiments, we use the AdamW optimizer ($\beta_1 = 0.95, \beta_2 = 0.999$) with a weight decay of $10^{-4}$. All models are trained for 300 epochs with a batch size of 64 and standard data augmentation on one NVIDIA RTX A6000 GPU. The number of modes, $K$, is set as a hyperparameter to follow prior work per task.

## 3    EXPERIMENTS

### 3.1    DATASETS

We evaluate our model on several standard benchmarks to cover a range of motion forecasting tasks. For 3D human pose prediction, we use Human3.6M (Ionescu et al., 2013), a large-scale lab-based dataset, and AMASS (Mahmood et al., 2019), a comprehensive motion capture archive used for generative modeling. For trajectory forecasting, we use the pedestrian datasets ETH-UCY (Lerner et al., 2007; Pellegrini et al., 2009) and the Stanford Drone Dataset (SDD) (Ro et al., 2019), which contains varied persons from an aerial view. Finally, we evaluate joint pose and trajectory prediction using Mocap-UMPM (CMU Graphics Lab, 2003; van der Aa et al., 2011), a mixed dataset of Mocap and UMPM containing synthesized human interaction between three people, and 3DPW (von Marcard et al., 2018), a dataset with two people traversing a real-world environment. We report results on each benchmark after training our model on its respective dataset in Table 1, which uses the same color scheme to visually group the results by task.

### 3.2    METRICS

We evaluate our model following common practice for multi-modal models that generate $K$ proposals, reporting the minimum error among all generated proposals. For pose prediction, we report the minimum Average/Final Displacement Error (ADE/FDE) averaged across all body joints over $K = 7$ proposals, following Hosseininejad et al. (2025). For trajectory prediction, we report the ADE/FDE on the root joint over $K = 20$ proposals, following Yao et al. (2024). In the combined pose and trajectory prediction task, we assess local and global accuracy over $K = 6$ proposals, following Jeong et al. (2024). For this, we use two metrics: Aligned mean per joint Position Error (APE), which measures pose error after root-alignment, and Joint Precision Error (JPE), which measures the overall error of all joints in the world coordinate system. Consistent with prior work, for datasets containing multiple people, the final reported metric is the average of the errors computed for each individual.

### 3.3    BASELINES

We compare our method against a wide range of state-of-the-art models across three distinct prediction tasks. In the domain of pose-only prediction, we evaluate against several recent generative approaches, including DivSamp (Dang et al., 2022), and prominent diffusion-based models such as BeLFusion (Barquero et al., 2023), CoMusion (Sun & Chowdhary, 2024), and SkeletonDiff (Curreli et al., 2025). Our comparison in this category also includes Motionmap (Hosseininejad et al., 2025) and the state-space diffusion model SLD (Xu et al., 2024). For trajectory-only prediction, we benchmark against MID (Gu et al., 2022), GP-Graph (Bae et al., 2022), TUTR (Shi et al., 2023), SingularTrajectory (Bae et al., 2024), the vision-language model TrajCLIP (Yao et al., 2024), and NMRF (Fang et al., 2025). Finally, for the comprehensive task of multi-person motion prediction, which involves forecasting combined human trajectory and pose, we include EMPMP (Zheng et al., 2025) and T2P (Jeong et al., 2024).

### 3.4    QUANTITATIVE RESULTS

Our proposed simple model demonstrates versatile and robust performance, improving state-of-the-art results across a diverse range of motion forecasting tasks, as shown in Table 1. Its success as a generalist architecture is particularly noteworthy given that many competing methods are

Table 1: Detailed comparison of model performance. Lower values are better (↓), with the best results shown in **bold**. An asterisk (*) denotes models we recomputed for this setup, a dagger (†) marks models adapted for the specific task, while a (∧) notes models that use external training data.

| | | Pose Prediction | | Trajectory Prediction | | Pose + Trajectory Prediction | |
|---|---|---|---|---|---|---|---|
| | **Dataset**
**In/Out length (s)**
**Metric** | **Human3.6M**
**0.5/2.0**
ADE↓/FDE↓ | **AMASS**
**0.5/2.0**
ADE↓/FDE↓ | **ETH-UCY (Avg)**
**3.2/4.8**
ADE↓/FDE↓ | **SDD**
**3.2/4.8**
ADE↓/FDE↓ | **MOCAP-UMPM**
**1.0/2.0**
APE↓/JPE↓ | **3DPW**
**0.8/1.6**
APE↓/JPE↓ |
| **Pose** | DivSamp | 0.48/0.68 | 0.48/0.64 | - | - | - | - |
| | BeLFusion | 0.44/0.60 | 0.35/0.48 | - | - | - | - |
| | CoMusion | 0.43/0.61 | 0.31/0.46 | - | - | - | - |
| | Motionmap | 0.47/0.60 | 0.32/0.45 | - | - | - | - |
| | SkeletonDiff | 0.64/0.77* | 0.56/0.71* | - | - | - | - |
| | SLD | **0.42**/0.59* | **0.30**/0.45* | - | - | - | - |
| **Traj** | MID | - | - | 0.21/0.38 | 7.61/14.32 | - | - |
| | GP-Graph | - | - | 0.23/0.39 | 9.10/13.76 | - | - |
| | TUTR | - | - | 0.21/0.36 | 7.76/12.69 | - | - |
| | SingularTrajectory | - | - | 0.22/0.34 | 7.26/12.58 | - | - |
| | TrajCLIP | - | - | **0.18**/0.33^ | 6.29/11.79^ | - | - |
| | NMRF | - | - | 0.19/**0.32** | 7.20/11.29 | - | - |
| **Pose+Traj** | T2P | 0.80/1.03† | 0.63/0.94† | 0.19/0.39† | 8.11/8.59† | 151.71/262.73 | 150.04/236.24 |
| | EMPMP | 0.45/0.72† | 0.42/0.65† | 0.63/0.72† | 10.29/10.51† | 146.52/250.41* | 150.62/235.44* |
| | Ours (wide) | **0.42**/0.59 | 0.31/**0.45** | **0.18**/**0.32** | 6.70/7.63 | **125.70**/212.72 | **142.89**/230.97 |
| | Ours (deep) | 0.44/**0.57** | 0.35/0.47 | 0.19/**0.32** | **6.26**/**7.61** | 131.41/**211.76** | 148.91/231.48 |

highly specialized and incorporate sophisticated, domain-specific inductive biases. For instance, top-performing baselines often rely on complex operations such as the Discrete Cosine Transform (DCT) to model motion in the frequency domain (Xu et al., 2024) or employ graph convolutional networks (GCNs) to explicitly encode the body's kinematic structure (Sun & Chowdhary, 2024). The results for our two primary configurations—a "wide" model and a "deep" model—highlight the effectiveness of our simple, unified approach in challenging established, task-specific methods.

On the Human3.6M benchmark, our model's performance matches the leading methods in Average Displacement Error (ADE) while outperforming compared methods in Final Displacement Error (FDE). This strength in long-term forecasting is further confirmed on AMASS, where it again surpasses existing models on the FDE metric. This success illustrates that attention-based transformers can effectively and accurately model high-dimensional pose data. Notably, our model achieves this performance in a single, deterministic forward pass. This differs from the iterative sampling process required for inference by leading generative models (Curreli et al., 2025; Sun & Chowdhary, 2024).

On trajectory prediction, our "wide" model's performance is on par with the current best techniques, matching the leading results on both ADE and FDE metrics for the ETH-UCY dataset, with a detailed breakdown of the individual ETH components available in the appendix. Given that these scenes can contain up to 57 pedestrians, our model's success is particularly notable, as it challenges the conventional wisdom that highly complex components are required for navigating crowded environments. For instance, our simple transformer architecture does not rely on the external knowledge of massive, pre-trained vision-language models as in TrajCLIP, or the continuous, field-based scene representations used by NMRF. Furthermore, on the SDD benchmark, both of our models outperform the prior work, with our deep configuration improving on FDE by 32%.

In the comprehensive task of combined pose and trajectory prediction, the advantages of our unified architecture are most prominent. On both the MOCAP-UMPM and 3DPW datasets, our models substantially outperform prior methods like T2P and EMPMP. These competing approaches often rely on complex, multi-stage pipelines, where localized and global aspects of motion are processed during separate stages (Jeong et al., 2024). In contrast, by jointly modeling pose and trajectory within a single end-to-end framework, our approach more effectively captures the coupled dynamics between local body articulation and global root movement, leading to significant performance gains across all metrics. For instance, on MOCAP-UMPM, our models lower the APE by more than 10.3% and JPE by 15%.

Our model's strong performance on multi-person datasets is achieved without any explicit interaction modules, since we treat any individuals in a scene independently. The success stems from our powerful single-agent motion representation, which not only validates the foundational architecture

but also reveals a clear opportunity for future work: integrating an explicit interaction mechanism could yield even better results.

Additionally, we want to note that our model is computationally very efficient. To demonstrate this, we benchmarked all models that perform joint pose and trajectory prediction on the MOCAP-UMPM dataset, comparing the average number of samples processed per second. Our "deep" configuration is not only more accurate but also more computationally efficient than the lightweight EMPMP model, showing a 14.3% increase in training throughput and processing test samples nearly 1.8 times faster. Please see Table 2 for details.

Table 2: Throughput mean $\pm$ std calculated over 10 runs on MOCAP-UMPM data. All models are run on a NVIDIA RTX A6000 GPU with batch size 64. Higher values are better ($\uparrow$). An asterisk(*) denotes models we recomputed for this experiment.

| Model | Training Throughput (samples/sec) $\uparrow$ | Test Throughput (samples/sec) $\uparrow$ |
|---|---|---|
| T2P* | $187 \pm 22$ | $401 \pm 64$ |
| EMPMP* | $812 \pm 58$ | $2041 \pm 129$ |
| Ours (wide) | $862 \pm 43$ | $2251 \pm 140$ |
| Ours (deep) | $\mathbf{928} \pm 45$ | $\mathbf{3673} \pm 161$ |

### 3.5 QUALITATIVE RESULTS

We provide a qualitative comparison of predicted motions on the MOCAP-UMPM dataset in Figure 2. The figure illustrates a challenging sample where three individuals are walking backward, a motion that requires complex coordination. Our "wide" and "deep" models both generate fluid and physically plausible motion sequences that accurately capture the underlying dynamics. The articulation of the arms and torso is notably realistic, showcasing the model's ability to learn natural human motion without being constrained by explicit structural priors. In particular, our "deep" configuration demonstrates exceptional performance over the long term, maintaining high-quality, dynamic predictions even at the final $t = 2.0s$ timestep.

The performance of the baseline models highlights the advantages of our unified approach. T2P resorts to an overly conservative strategy when challenged with this tricky, high-uncertainty scenario. Its predictions become increasingly static over time, collapsing towards a mean pose with very little movement to avoid large errors. In contrast, EMPMP attempts to generate dynamic motion but struggles with physical plausibility. Its predictions exhibit noticeable artifacts, such as the unnatural arm posture of the person in green and the awkward leg movements of the person in blue. These qualitative results underscore that our model not only achieves superior quantitative accuracy but also produces motions that are significantly more realistic and coherent than competing methods.

Figure 2: Visualization of predictions on a MOCAP-UMPM scene. Model predictions are in color, and ground truth future poses are black dashes. The last-known input positions are colored dashes.

Table 3: Comparison of our model's performance with different hyperparameter configurations.

| Depth | Embed dim | Total Params | APE | JPE |
|-------|-----------|--------------|-----|-----|
| 8 | 192 | 5.2M | 126.05 | 212.84 |
| 6 | 192 | 4.0M | **125.70** | 212.72 |
| 4 | 192 | 2.8M | 126.22 | 213.40 |
| 12 | 96 | 1.9M | 128.47 | 212.08 |
| 6 | 96 | 1.0M | 128.52 | 212.45 |
| 12 | 64 | 860K | 130.72 | 212.30 |
| 16 | 48 | 642K | 131.41 | **211.76** |
| 12 | 48 | 490K | 131.09 | 212.73 |
| 16 | 36 | 367K | 134.52 | 215.36 |

## 3.6 ABLATION STUDIES

In this section, we conduct a series of ablation studies to investigate the impact of our model's key components and hyperparameters. We perform these experiments on the MOCAP-UMPM dataset for the joint pose and trajectory prediction task to analyze the effectiveness of our multi-modal prediction head and the trade-offs in our transformer architecture.

### 3.6.1 CHOICE OF TRANSFORMER HYPERPARAMETERS

Our model's major computations are performed using a simple transformer decoder. We analyze the trade-offs between its depth (number of layers, $L$) and width (embedding dimension, $d_{model}$). We experimented with various configurations, keeping the overall parameter count relatively low, to find effective deep net architecture designs. The results are summarized in Table 3.

The analysis reveals a clear relationship between depth, width, and predictive accuracy. Our "wide" configuration ($L = 6, d_{model} = 192$) achieves the best APE, suggesting that a more expansive embedding space is beneficial for capturing fine-grained pose details. Decreasing the depth to $L = 4$ or increasing it to $L = 8$ with the same width leads to a decline in performance, indicating a sweet spot for this configuration.

Conversely, our "deep" model ($L = 16, d_{model} = 48$) obtains the lowest JPE, demonstrating that a deeper stack of attention layers is more effective at modeling complex, long-range spatio-temporal dependencies for global trajectory prediction, even with a constrained embedding dimension. As expected, performance degrades significantly with shallower or narrower architectures. These results validate our choice of the "wide" and "deep" models, as they represent two distinct and highly effective points in the architecture design space, tailored for different aspects of motion prediction.

### 3.6.2 EFFECT OF MULTI-MODAL PREDICTION

While multi-modal prediction is standard in trajectory forecasting, state-of-the-art methods for joint pose and trajectory prediction, such as EMPMP, have often favored a deterministic approach, predicting a single future outcome. However, human motion is inherently stochastic, and a single prediction can fail to capture the full range of plausible futures. We therefore conduct an ablation to quantify the advantage of our multi-modal prediction head explicitly. We compare our model's performance when generating multiple proposals ($K = 6$) against a deterministic setting ($K = 1$), mirroring the setup of prior work (Jeong et al., 2024).

The results in Table 4 clearly demonstrate the limitations of a deterministic approach. Even in a deterministic setting, our "wide" model is already competitive with EMPMP. However, by embracing multi-modality, our model achieves a dramatic performance gain. The APE improves by 13.8% and the JPE by a substantial 24.2%. This highlights that our model doesn't just produce a better single guess; it effectively

Table 4: Model performance with 2 different modes on MOCAP-UMPM data. Lower values are better ($\downarrow$).

| Model | $K = 1$ | | $K = 6$ | |
|-------|-----|-----|-----|-----|
| | **APE** | **JPE** | **APE** | **JPE** |
| T2P | 154.4 | 366.4 | 151.7 | 262.7 |
| EMPMP | 147.2 | 283.1 | 146.5 | 250.4 |
| Ours (wide) | **145.84** | **280.8** | **125.70** | 212.72 |
| Ours (deep) | 149.35 | 286.96 | 131.41 | **211.76** |

captures a distribution of high-quality future motions. Interestingly, prior works do not benefit from multiple modes to the same degree. For instance, EMPMP's APE barely improves, suggesting its architecture may struggle to generate genuinely distinct futures. While a full analysis of why the baselines are less suited to multi-modal prediction is beyond the scope of this paper, it suggests that our unified architecture is particularly effective at leveraging the "winner-takes-all" loss to produce a diverse and plausible set of outcomes—a crucial capability that deterministic models lack by design.

# 4 RELATED WORK

Human motion requires a holistic assessment, as local body articulation (pose) and global displacement (trajectory) are deeply intertwined. Our research community, however, has largely tackled motion prediction by decomposing this process into specialized sub-problems: pose, trajectory, and multi-person motion prediction. This specialization has driven progress on narrow benchmarks but created a dichotomy: specialized models fail to generalize, while the few holistic models struggle to compete on established task-specific leaderboards. This "benchmark effect" has incentivized an escalation in architectural complexity, with increasingly elaborate models gaining an edge.

In this context, the transformer has emerged as a powerful tool for sequence modeling. However, its application to human motion has often followed the increasing complexity trend, where it merely serves as a backbone for other domain-specific modules. This paper challenges that approach. We posit that the transformer's true power lies not in its ability to support additional complex components, but in its inherent capacity to address the problem in a simple, direct, and unified manner.

## 4.1 HUMAN POSE PREDICTION

The task of human pose prediction involves forecasting a future sequence of 3D skeletal joint locations relative to the root joint based on an observed history of poses (Hosseininejad et al., 2025, see Appendix C). To address the stochastic nature of human behavior, the field has shifted from deterministic models (Medjaouri & Desai, 2022; Ma et al., 2022) to complex generative frameworks, particularly diffusion models. This pursuit of generative fidelity has fueled a cycle of escalating complexity, with methods like BeLFusion (Barquero et al., 2023) introducing a "behavioral latent space" and CoMusion (Sun & Chowdhary, 2024) employing a hybrid Transformer-GCN architecture that operates in the Discrete Cosine Transform (DCT) (Mao et al., 2021) space to model skeletal kinematics explicitly. Recent methods like SkeletonDiff (Curreli et al., 2025) and SLD (Xu et al., 2024) focus on skeleton-aware generation or long-sequence efficiency, while non-diffusion approaches like Motionmap (Hosseininejad et al., 2025) introduce novelties such as multi-stage heatmap pipelines.

## 4.2 HUMAN TRAJECTORY PREDICTION

Trajectory forecasting aims to predict the future path of an agent's root joint, a task complicated by latent intent, social interactions, and environmental constraints. Recent state-of-the-art approaches have often relied on massive external knowledge sources or engineered, multi-stage pipelines. A prominent trend involves leveraging large foundation models; TrajCLIP (Yao et al., 2024), for example, incorporates knowledge from vision-language models (VLMs) to provide contextual cues, effectively outsourcing the learning problem. Another approach involves building complex frameworks for generality, such as Singular Trajectory (Bae et al., 2024), whose "universal" status is the result of an engineered pipeline involving Singular Value Decomposition and a diffusion-based refiner, or NMRF (Fang et al., 2025), which uses sophisticated modules like continuous, field-based scene representations.

## 4.3 COMBINED POSE AND TRAJECTORY PREDICTION

The simultaneous prediction of pose and trajectory is where the limitations of fragmented architectures become most apparent, as this task requires modeling the critical coupling between local articulation and global movement. Early work such as Tripod (Adeli et al., 2021) and work by Zaier et al. (2023) established the importance of forecasting these dynamics jointly, typically employing graph-based or multi-branch architectures to capture the dependencies. More recent approaches have explored pre-training strategies, such as Multi-transmotion (Gao et al., 2024), to learn general-

izable motion representations. Despite this progress, prior~~Prior~~ work has typically imposed strong architectural priors on how pose and trajectory information should interact. T2P (Jeong et al., 2024) employs a sequential, "coarse-to-fine" strategy, first predicting the global trajectory and then conditioning the pose prediction on that result. This design imposes a one-way causal assumption that trajectory dictates pose and is susceptible to error propagation. An alternative, seen in EMPMP (Zheng et al., 2025), uses parallel branches to process local and global information separately before fusion. This avoids direct error propagation but imposes a prior: that local and global features are separable concerns. This rigid separation may preclude the model from learning more complex, deeply intertwined representations where local and global dynamics are jointly encoded from the outset. Consequently, although EMPMP was explicitly designed to be "lightweight", its architecture is built from individually light but intricately integrated components and struggles to leverage hardware parallelism effectively.

## 5 CONCLUSION

This paper introduces SimpliHuMoN, a simple and unified transformer-based model that addresses the prevailing trends of fragmentation and escalating complexity in human motion prediction. We challenge the field by demonstrating how a single, end-to-end framework effectively learns the dynamics of human movement across various tasks. Extensive experiments across a wide range of standard benchmarks validate this approach, showing that our model achieves state-of-the-art accuracy while also proving more computationally efficient than prior methods. Ultimately, this work serves as evidence that architectural simplicity, when thoughtfully applied, can outperform engineered complexity, suggesting that the path forward in motion prediction lies not in adding more intricate components but in refining simple and truly generalizable foundations.

## 6 REPRODUCIBILITY STATEMENT

We are committed to ensuring the reproducibility of our research. Our model's architecture, loss function, training procedure, and key hyperparameters are described in detail in Section 2 of the main paper, with further analysis in our ablation studies (Section 3.6). For data handling, Appendix A provides a complete description of all datasets and the exact preprocessing steps, which follow established protocols from prior work. The precise mathematical definitions for all evaluation metrics are detailed in Appendix B. Finally, our supplementary website, referenced in Appendix C, offers additional qualitative results. Collectively, these resources provide a comprehensive guide for reproducing our experimental findings. We will also release all code.

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

APPENDIX: SIMPLIHUMON: SIMPLIFYING HUMAN MOTION PREDICTION

This appendix is structured as follows: In Sec. A we provide additional dataset and metric details. In Sec. B we detail additional experimental results. In Sec. C we highlight the website which is part of the provided appendix. In Sec. D we discuss joint training results. In Sec. E we provide information about our LLM usage.

## A  ADDITIONAL DATASET AND METRIC DETAILS

### A.1  SOURCES AND PROCESSING OF DATA

All experiments are conducted on publicly available, open-source datasets. To ensure a fair and direct comparison with prior work, we strictly adhere to the established data processing and evaluation protocols from recent top-performing methods for each prediction task. This standardization ensures that the performance improvements reported in this paper are attributable to our model's architecture rather than differences in data handling. The specific protocols are as follows: For pose prediction on the Human3.6M and AMASS datasets, we follow the data processing methodology, sequence lengths, and evaluation splits established by BeLFusion (Barquero et al., 2023). For trajectory prediction on the ETH-UCY and SDD, our data handling and evaluation procedures align with the protocol set forth by NMRF (Fang et al., 2025). For combined pose and trajectory Prediction on the MOCAP-UMPM and 3DPW datasets, we adopt the data preparation and processing pipeline outlined by T2P (Jeong et al., 2024).

### A.2  METRIC FORMULAE

Given the predicted motion proposal $X_{\text{fut}}^k = \{x_{t,m}^k\} \in \mathbb{R}^{F \times M \times 3}$ for $k \in \{1, 2, ..., K\}$ across $F$ time frames with $M$ joints per person, along with the corresponding ground truth $X_{\text{fut}}^{\text{gt}} = \{x_{t,m}^{\text{gt}}\}$, the following metrics are used for evaluation. For multi-modal predictions, we follow common practice and report the minimum error among all $K$ generated proposals for each metric (*e.g.*, minADE, minFDE). Consistent with prior work, for datasets containing multiple people, the final reported error is the average of the metric computed for all individuals. All metrics in the main paper are reported for the final output timestep, $t = F$.

**APE.**  Aligned mean per joint Position Error (APE) is used as a metric to evaluate the forecasted local motion. Euclidean distance of each joint relative to the root (hip) joint is averaged over all joints for a given timestep, $t$:

$$\text{APE}_t(X_{\text{fut}}^{\text{gt}}, X_{\text{fut}}^k) = \frac{1}{M} \sum_{m=1}^{M} \|(x_{t,m}^{\text{gt}} - x_{t,\text{hip}}^{\text{gt}}) - (x_{t,m}^k - x_{t,\text{hip}}^k)\|_2. \tag{2}$$

**JPE.**  Joint Precision Error (JPE) evaluates both global and local predictions by the average Euclidean distance of all joints for a given timestep, $t$:

$$\text{JPE}_t(X_{\text{fut}}^{\text{gt}}, X_{\text{fut}}^k) = \frac{1}{M} \sum_{m=1}^{M} \|x_{t,m}^{\text{gt}} - x_{t,m}^k\|_2. \tag{3}$$

**ADE.**  Average Displacement Error (ADE) measures the Euclidean distance between the ground truth and predicted sequences, averaged over all joints and all future time frames:

$$\text{ADE}(X_{\text{fut}}^{\text{gt}}, X_{\text{fut}}^k) = \frac{1}{F \times M} \sum_{t=1}^{F} \sum_{m=1}^{M} \|x_{t,m}^{\text{gt}} - x_{t,m}^k\|_2. \tag{4}$$

**FDE.**  Final Displacement Error (FDE) measures the Euclidean distance between the ground truth and the prediction, averaged over all joints for a given timestep, $t$:

$$\text{FDE}_t(X_{\text{fut}}^{\text{gt}}, X_{\text{fut}}^k) = \frac{1}{M} \sum_{m=1}^{M} \|x_{t,m}^{\text{gt}} - x_{t,m}^k\|_2. \tag{5}$$

Table 5: Trajectory prediction performance (ADE/FDE) on ETH-UCY. Lower values are better, with the best results shown in **bold**. A dagger (†) marks models adapted for the specific task, while a (∧) notes models that use external training data.

| Model | ETH | HOTEL | UNIV | ZARA1 | ZARA2 | AVG |
|---|---|---|---|---|---|---|
| MID | 0.39/0.66 | 0.13/0.22 | 0.22/0.45 | 0.17/0.30 | 0.13/0.27 | 0.21/0.38 |
| GP-Graph | 0.43/0.63 | 0.18/0.30 | 0.24/0.42 | 0.17/0.31 | 0.15/0.29 | 0.23/0.39 |
| TUTR | 0.40/0.61 | 0.11/0.18 | 0.23/0.42 | 0.18/0.34 | 0.13/0.25 | 0.21/0.36 |
| SingularTrajectory | 0.35/0.42 | 0.13/0.19 | 0.25/0.44 | 0.19/0.32 | 0.15/0.25 | 0.22/0.34 |
| TrajCLIP$^\wedge$ | 0.36/0.57 | **0.10/0.17** | **0.19/0.41** | **0.16/0.28** | **0.11/0.20** | **0.18**/0.33 |
| NMRF | **0.26/0.37** | 0.11/0.17 | 0.28/0.49 | 0.17/0.30 | 0.14/0.25 | 0.19/**0.32** |
| T2P$^\dagger$ | 0.29/0.55 | 0.15/0.27 | 0.25/0.53 | **0.16**/0.33 | 0.12/0.26 | 0.19/0.39 |
| EMPMP$^\dagger$ | 0.99/0.98 | 0.70/0.87 | 0.69/0.89 | 0.43/0.50 | 0.32/0.35 | 0.63/0.72 |
| Ours (wide) | 0.28/0.44 | 0.13/0.24 | 0.24/0.44 | **0.16**/0.29 | **0.11**/0.21 | **0.18/0.32** |
| Ours (deep) | 0.29/0.44 | 0.14/0.24 | 0.24/0.43 | 0.17/0.29 | 0.13/0.21 | 0.19/**0.32** |

## B  ADDITIONAL EXPERIMENTAL RESULTS

### B.1  PER-DATASET SPLIT ON ETH-UCY

On the ETH-UCY datasets, our model demonstrates highly competitive performance against leading methods, as detailed in Table 5. While models like TrajCLIP (Yao et al., 2024) and NMRF (Fang et al., 2025) achieve the best results on some of the individual scenes, our "wide" configuration achieves the best overall performance, tying for the best average ADE (0.18) and the best average FDE (0.32).

This result is particularly noteworthy when considering the architectural differences between our model and methods like TrajCLIP. TrajCLIP's strong performance stems from its use of a large, pre-trained VLM to provide rich semantic priors. Specifically, it uses natural language prompts (*e.g.*, "a person walking") to generate contextual embeddings from the VLM's text encoder, which are then fused with visual features to guide the trajectory prediction. This approach effectively outsources a part of the learning problem to a massive external knowledge base. While powerful, this creates a dependency on computationally heavy external models and assumes that general web-scale knowledge is optimally suited for the fine-grained physics of trajectory prediction.

Our model, in contrast, is entirely self-contained, learning all necessary dynamics exclusively from the provided motion data. The performance difference on the ETH scene, where our model significantly outperforms TrajCLIP, suggests a key advantage of this self-sufficient approach. The ETH dataset represents a scenario where the visual-semantic cues that TrajCLIP relies on are less informative and reliable than in other scenes. In such cases, our model's ability to learn robustly from the motion dynamics alone allows it to generalize more effectively, leading to a more consistent performance profile across all five datasets. This consistency is what enables our model to achieve better average performance without relying on external priors, challenging the notion that they are a prerequisite for top-tier trajectory forecasting.

Furthermore, a key architectural difference is TrajCLIP's explicit modeling of social and environmental interactions through two dedicated modules. They are designed to capture the dynamics between different agents and integrate visual context from the environment to make predictions physically consistent with the static scene. In contrast, our current model processes each agent independently and contains no such explicit interaction mechanisms. The fact that our simpler, non-interactive approach still achieves state-of-the-art average performance highlights the remarkable strength and efficiency of its core motion representation. This also points to a promising avenue for future work: integrating a lightweight interaction mechanism into our powerful architecture could potentially push performance even further.

Table 6: Comparison of APE/JPE metrics across models and datasets. Lower values are better ($\downarrow$), with the best results shown in **bold**. An asterisk (*) denotes models we recomputed for this setup.

| | | MOCAP-UMPM | | | | | 3DPW | | | |
|---|---|---|---|---|---|---|---|---|---|---|
| | In/Out Length (s) | 0.4s | 0.8s | 1.2s | 1.6s | 2.0s | 0.4s | 0.8s | 1.2s | 1.6s |
| **APE** | T2P* | 71.7 | 107.8 | 120.4 | 137.1 | 151.7 | 98.2 | 114.6 | 135.3 | 150.0 |
| | EMPMP* | 60.1 | 96.0 | 116.9 | 131.6 | 146.5 | 96.3 | 111.9 | 134.4 | 150.6 |
| | Ours (wide) | **57.3** | **87.7** | **104.5** | **115.3** | **125.7** | 92.8 | **107.1** | **130.0** | **142.9** |
| | Ours (deep) | 62.3 | 89.5 | 107.3 | 119.0 | 128.5 | **93.2** | 108.4 | 131.5 | 148.9 |
| **JPE** | T2P* | 70.2 | 139.2 | 160.1 | 226.4 | 262.7 | 107.7 | 142.6 | 181.0 | 236.2 |
| | EMPMP* | 68.0 | 123.9 | 170.3 | 219.1 | 250.4 | 103.6 | 140.2 | 179.8 | 235.4 |
| | Ours (wide) | **64.6** | **108.6** | **143.9** | 177.7 | 212.7 | **99.4** | **137.3** | 172.1 | **231.0** |
| | Ours (deep) | 68.9 | 109.9 | 145.3 | **177.2** | **210.3** | 100.1 | 138.2 | **171.6** | 231.5 |

## B.2 DETAILED METRICS ACROSS KEY FRAMES

To scrutinize performance over the forecast horizon, Table 6 presents a time-step-level analysis on the MOCAP-UMPM and 3DPW datasets. The results reveal not only the consistent superiority of our models over T2P and EMPMP at every interval but also a crucial architectural trade-off.

Our "wide" model establishes a new standard for local pose accuracy (APE), excelling at capturing fine-grained kinematics, particularly in the short term. Conversely, our "deep" model demonstrates its strength in long-range forecasting, achieving the best overall world-coordinate accuracy (JPE) at the final timesteps. This divergence highlights a key finding: architectural depth appears more critical for maintaining global trajectory coherence, while width is more effective for local pose detail. Most notably, the performance gap between our models and the baselines widens as the prediction horizon increases. This demonstrates our architecture's superior robustness against the error accumulation that typically plagues sequential prediction tasks. This detailed analysis confirms that our simple, unified framework is not just more accurate overall but is also more effective at handling the challenges of long-term motion forecasting compared to competing multi-stage or specialized approaches.

## C WEBSITE

We provide a website with additional visualizations demonstrating our method's performance, which can be accessed using the provided HTML file.

We observe that the generated motions exhibit high physical plausibility, with no unrealistic artifacts such as foot sliding. Body poses are consistently realistic, respecting natural body constraints and capturing fine-grained details without grouping different joints into unnatural, blocky movements. Furthermore, our model adeptly handles both independent and coupled motion dynamics; it accurately predicts localized movements (*e.g.*, arm gestures without a change in trajectory) and complex actions where limb articulation and global trajectory are deeply intertwined. Our model excels in multi-person scenes by processing agents independently. This avoids a key limitation of rigid, graph-based interaction models (GNNs), which can corrupt individual forecasts by forcing information aggregation from non-interacting neighbors. This finding does not diminish the importance of interaction modeling but rather clarifies the need to learn it dynamically.

## D JOINT TRAINING

To test the full generalization capability of our architecture, we train a single, universal model jointly on all datasets across all tasks (pose, trajectory, and combined prediction). This experiment aims to create a single set of weights that can perform any of the specialized tasks without retraining. Handling the significant diversity in data formats, skeleton structures, and sequence lengths requires a carefully designed methodology, which we detail below.

Table 7: Mapping from dataset-specific skeletons to our 22-joint canonical representation. AMASS serves as the canonical skeleton itself. Dashes (–) indicate that a direct mapping for that specific canonical joint is unavailable in the source dataset.

| # | AMASS | Human3.6M | MOCAP-UMPM | 3DPW |
|---|---|---|---|---|
| 1 | Pelvis | – | Hips | Pelvis |
| 2 | L_Hip | LeftUpLeg | LHip | LHip |
| 3 | R_Hip | RightUpLeg | RHip | RHip |
| 4 | Spine1 | Spine | Spine | – |
| 5 | L_Knee | LeftLeg | LKnee | LKnee |
| 6 | R_Knee | RightLeg | RKnee | RKnee |
| 7 | Spine2 | – | – | – |
| 8 | L_Ankle | LeftFoot | LAnkle | – |
| 9 | R_Ankle | RightFoot | RAnkle | – |
| 10 | Spine3 | – | – | – |
| 11 | L_Foot | – | – | LFoot |
| 12 | R_Foot | – | – | RFoot |
| 13 | Neck | Neck | Neck | – |
| 14 | L_Collar | – | – | – |
| 15 | R_Collar | – | – | – |
| 16 | Head | Head / Head-top | Head | – |
| 17 | L_Shoulder | LeftArm | LShoulder | LShoulder |
| 18 | R_Shoulder | RightArm | RShoulder | RShoulder |
| 19 | L_Elbow | LeftForeArm | LElbow | LElbow |
| 20 | R_Elbow | RightForeArm | RElbow | RElbow |
| 21 | L_Wrist | LeftHand | LWrist | LWrist |
| 22 | R_Wrist | RightHand | – | RWrist |

## D.1 METHODOLOGY

**Data Unification and Canonical Skeleton.** A primary challenge is the heterogeneity of the datasets. To create a consistent input format, all data is preprocessed into a normalized tensor of shape $T \times M \times 3$ (sequence length $\times$ joints $\times$ coordinates). We pad the data with a zero Z-dimension for 2D trajectory datasets (ETH-UCY, SDD) to create a consistent 3D representation.

To address the varying skeleton definitions, we establish a 22-joint canonical skeleton, using the AMASS dataset as our standard. All other datasets are mapped to this representation, as shown in Table 7. This mapping allows us to use a fixed set of learnable joint embeddings, ensuring that input data for a given semantic body part (*e.g.*, the 'Left Knee') is always processed by its corresponding embedding, regardless of the source dataset. For trajectory-only datasets, the single trajectory point is mapped to the 'Pelvis' joint embedding.

**Dataset-Balanced Batching.** We employ a dataset-balanced batching strategy to prevent the model from overfitting to larger datasets (*e.g.*, AMASS). Each training batch contains samples drawn from only a single dataset. We iterate through an equal number of batches from every dataset during each epoch, ensuring the model is exposed to a balanced distribution of tasks and data sources during training.

**Task-Specific Processing.** We use a task-type flag associated with each dataset to direct samples through the appropriate processing pipelines. For instance, a 'trajectory' flag ensures that data only passes through the trajectory-related input and output heads of the model, while a 'joint' flag activates both pose and trajectory heads. This allows the shared transformer core to learn a general motion representation while the specialized heads handle the task-specific details.

**Unified Model with Dynamic Slicing.** The model's internal parameters are defined by the maximum sequence length, $\max(T)$, and maximum number of joints, $\max(M)$, across all datasets. However, at runtime, a given sample's input and output tensors are dynamically sliced to match the

Table 8: Comparison of performance on individual vs. joint training. Lower values are better (↓).

| Dataset
In/Out length (s)
Metric | Pose Prediction | | Trajectory Prediction | | Pose + Trajectory Prediction | |
|---|---|---|---|---|---|---|
| | Human3.6M
0.5/2.0
ADE↓/FDE↓ | AMASS
0.5/2.0
ADE↓/FDE↓ | ETH-UCY (Avg)
3.2/4.8
ADE↓/FDE↓ | SDD
3.2/4.8
ADE↓/FDE↓ | MOCAP-UMPM
1.0/2.0
APE↓/JPE↓ | 3DPW
0.8/1.6
APE↓/JPE↓ |
| Ours (wide, ind.) | 0.42/0.59 | 0.31/0.45 | 0.18/0.32 | 6.70/7.63 | 125.70/212.72 | 142.89/230.97 |
| Ours (deep, ind.) | 0.44/0.57 | 0.35/0.47 | 0.19/0.32 | 6.26/7.61 | 131.41/211.76 | 148.91/231.48 |
| Ours (wide, joint) | 0.49/0.63 | 0.51/0.66 | 0.23/0.37 | 9.04/11.21 | 135.19/220.13 | 150.40/234.81 |
| Ours (deep, joint) | 0.55/0.70 | 0.62/0.78 | 0.25/0.39 | 10.66/12.14 | 138.20/223.49 | 151.46/235.05 |

specific $T$ and $M$ of its source dataset. This allows a single, fixed-size model to efficiently process variable-dimension inputs and outputs.

## D.2 RESULTS

The results of our joint training experiment, presented in Table 8, demonstrate both the promise and the challenges of creating a single, universal motion prediction model. As expected, there is a performance trade-off when compared to the specialized models trained on individual datasets. The jointly trained models exhibit a degradation in accuracy across all tasks and datasets. However, the degree of this degradation varies, providing valuable insights into the model's behavior.

The "wide" model consistently outperforms the "deep" model in the joint training setting. This is the inverse of our findings in some specialized tasks, and it suggests that the higher parameter count and wider embedding dimension of the "wide" model provide the necessary capacity to learn a shared representation across the seven diverse datasets. The "deep" model, with its constrained architecture, likely lacks the capacity to effectively generalize across such a heterogeneous data distribution, leading to a more significant performance drop. We also observe that the performance degradation is most pronounced on the AMASS dataset. This is likely a direct consequence of our dataset-balanced batching strategy. While this strategy prevents the model from overfitting to the largest datasets, it also means that the model is significantly under-exposed to the vast and diverse AMASS dataset, which is over 140 times larger than the smallest dataset (SDD). The model simply does not see enough of the AMASS data distribution to learn it as effectively as the specialized model.

Despite the performance trade-off, these results represent a successful proof of concept. The ability of a single, simple architecture to perform pose prediction, trajectory forecasting, and combined holistic prediction without any architectural changes is a powerful demonstration of its inherent generality. The fact that the model produces reasonable, albeit less accurate, predictions across all tasks indicates that it has learned a meaningful and transferable internal representation of human motion. This experiment validates the potential for developing true "foundation models for motion." While our current approach shows a performance gap, it highlights a clear and promising research direction. Future work could focus on more sophisticated data-balancing techniques, curriculum learning strategies, or simply scaling the model's capacity to bridge this gap. The ability to train a single model that understands the principles of human motion across myriad contexts remains a valuable and achievable goal for the field.

## E LLM USAGE

While preparing this work, we used an LLM to assist with language editing and code generation for LaTeX tables and visualizations. The LLM's contributions were limited to improving the clarity of the text and formatting results. The core research, experimental design, and all scientific claims remain our original work.

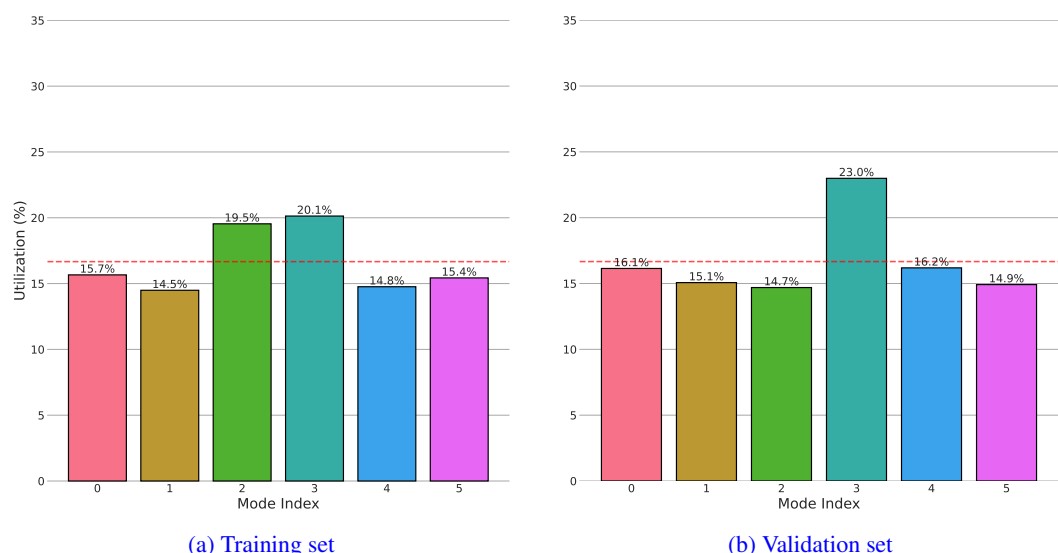

(a) Training set          (b) Validation set

Figure 3: Distribution of winning mode indices (best-of-6) on pose + trajectory prediction task across the training and validation sets of MOCAP-UMPM. The dashed line (- - -) indicates equal distribution. Both distributions demonstrate balanced mode utilization without mode collapse.

# F    EVALUATING PREDICTION DIVERSITY

A key component of our model is the multi-modal prediction head, which generates $K$ distinct hypotheses to account for the uncertain nature of human motion. To validate its effectiveness, we analyze two potential concerns: mode collapse and the true diversity of the generated futures.

## F.1    MODE UTILIZATION ANALYSIS

To quantitatively verify that our model uses its full predictive capacity, we logged the index of the best (lowest error) hypothesis for every sample in the MOCAP-UMPM training and validation sets ($K = 6$). Figure 3 plots the distribution of these winning indices. The results show that the model does not suffer from mode collapse. All six modes are actively utilized in both training and validation, with utilization rates clustering around the ideal uniform distribution (16.7%, shown as a dashed line). This confirms that the "winner-takes-all" loss, when applied to our architecture, successfully encourages the different proposals to specialize and cover distinct, plausible outcomes.

| $t = 0.4s$ | $t = 0.8s$ | $t = 1.2s$ | $t = 1.6s$ | $t = 2.0s$ |
|---|---|---|---|---|

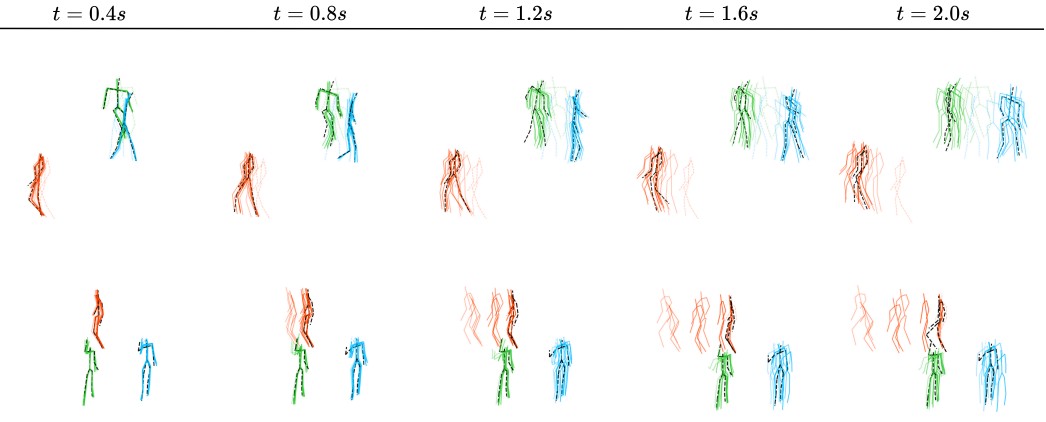

Figure 4: Visualization of motion proposals ($K = 6$) of our (wide) model on MOCAP-UMPM data. All model predictions are in color. Ground truth future poses are black dashes, and the last-known input positions are colored dashes.

Table 9: Comparison of pose prediction diversity. Lower values are better ($\downarrow$), with the best results shown in **bold**. An asterisk (*) denotes models we recomputed for this setup.

| Model | Type | Human3.6M | | AMASS | |
|---|---|---|---|---|---|
| | | MMADE↓ | MMFDE↓ | MMADE↓ | MMFDE↓ |
| DivSamp | Stochastic (Gumbel-Softmax) | 0.542 | 0.671 | 0.623 | 0.728 |
| BeLFusion | Stochastic (Latent Diffusion) | 0.491 | 0.586 | 0.488 | 0.564 |
| CoMusion | Stochastic (Motion Diffusion) | 0.531 | 0.623 | 0.526 | 0.602 |
| Motionmap | Stochastic (Multi-Stage Encoder-Decoder) | **0.466** | **0.532** | **0.450** | **0.514** |
| SkeletonDiff* | Stochastic (Gaussian Diffusion) | 0.568 | 0.694 | 0.641 | 0.740 |
| SLD* | Stochastic (State-Space Diffusion) | 0.497 | 0.576 | 0.482 | 0.551 |
| Ours (wide) | Deterministic (K-Proposal) | 0.526 | 0.587 | 0.519 | 0.560 |
| Ours (deep) | Deterministic (K-Proposal) | 0.535 | 0.597 | 0.521 | 0.571 |

### F.2 QUALITATIVE DIVERSITY VISUALIZATION

Figure 4 provides a qualitative visualization of this diversity for a sample from the MOCAP-UMPM dataset. The figure overlays all $K = 6$ proposals generated by our "wide" model. It clearly illustrates that the model is capturing distinct, high-level behaviors; for instance, the agent in red is predicted to either walk straight, stop, or turn, with each prediction representing a physically plausible and coherent motion.

### F.3 QUANTITATIVE DIVERSITY METRICS

To further assess the quality and diversity of our generated motion distribution, we report standard multimodal metrics, Minimum-over-K Average Displacement Error (MMADE) and Minimum-over-K Final Displacement Error (MMFDE), on the pose-only benchmarks. Table 9 compares our model's performance against prominent stochastic and generative baselines. Our deterministic K-proposal approach achieves MMADE/MMFDE scores that are competitive with these generative methods, demonstrating that our $K$ proposals capture a meaningful and diverse set of high-quality future motions.

## G ANALYSIS OF UNIFIED ARCHITECTURE

To validate our central claims, we present ablations addressing two critical questions. First, we provide quantitative proof that jointly modeling pose and trajectory is mutually beneficial. Second, we investigate why our simple, unified attention mechanism outperforms standard, more complex encoder-decoder designs. We also extend our ablation study from Section 3.6 to test the inclusion of some architectural components.

### G.1 BENEFIT OF JOINT MODELING

A core hypothesis of our work is that pose and trajectory are deeply intertwined and that modeling them jointly improves the prediction of both. We tested this hypothesis directly by training "pose-only" and "trajectory-only" variants of our model and comparing them to our full, joint model on MOCAP-UMPM.

Table 10 presents these results, providing strong quantitative evidence for our hypothesis. The data shows that pose prediction improves by $\sim$11-12% when trajectory is provided as an input, compared to a "pose-only" variant. Conversely, trajectory prediction improves by $\sim$12-14% when pose is provided as an input, compared to a "trajectory-only" variant. This confirms that our unified framework successfully learns the physical coupling between local articulation (pose) and global movement (trajectory), using information from one to refine its predictions for the other.

### G.2 UNIFIED SELF-ATTENTION VS. CROSS-ATTENTION

A key question is why our simple architecture works so well. We hypothesize it is due to our unified self-attention mechanism, where past context and future queries are concatenated as ($[\mathcal{C};$

Table 10: Comparison of the influence of combined pose and trajectory information vs. individual performance on MOCAP-UMPM data. Lower values are better ($\downarrow$), with the best results shown in **bold**. A dagger (†) marks models adapted for the specific task.

| Training Method Metric | Pose Prediction | | Trajectory Prediction | |
|---|---|---|---|---|
| | Only Pose ADE$\downarrow$/FDE$\downarrow$ | Pose + Traj ADE$\downarrow$/FDE$\downarrow$ | Only Traj ADE$\downarrow$/FDE$\downarrow$ | Pose + Traj ADE$\downarrow$/FDE$\downarrow$ |
| T2P† | - | 0.61/0.95 | - | 0.20/0.29 |
| EMPMP† | - | 0.53/0.65 | - | 0.14/0.20 |
| Ours (wide) | 0.46/0.58 | **0.41/0.51** | 0.09/0.18 | **0.08/0.16** |
| Ours (deep) | 0.46/0.59 | 0.42/**0.51** | 0.09/0.18 | **0.08**/0.17 |
| Improvement (%) | | -11.4/-12.2 | | -13.7/-11.9 |

Table 11: Ablation study on attention mechanisms measuring metrics on MOCAP-UMPM. Lower values are better ($\downarrow$), with the best results shown in **bold**.

| Attention Mechanism | Wide | | Deep | |
|---|---|---|---|---|
| | APE$\downarrow$ | JPE$\downarrow$ | APE$\downarrow$ | JPE$\downarrow$ |
| Self-Attn over $[\mathcal{C}; \mathcal{Q}]$ (Ours) | **125.70** | **212.72** | **131.41** | **211.76** |
| Self-Attn ($\mathcal{C}$) + Cross ($\mathcal{Q}$) | 134.61 | 229.05 | 140.32 | 227.89 |

$\mathcal{Q}$]) and processed in a single attention block. This differs from standard encoder-decoders that use separate self-attention on the context and cross-attention for the queries. To test this, we built a new baseline that replaces our unified attention with a standard encoder-decoder design, keeping all other parameters and hyperparameters identical.

Table 11 shows the results. Our unified Self-Attn($[\mathcal{C}; \mathcal{Q}]$) mechanism outperforms the standard cross-attention baseline, improving APE by 6.6% and JPE by 7.1% for the wide model. We believe this is because unified attention allows for a richer, bidirectional information flow at every step.

Figure 5 provides a visualization of the attention patterns in our first transformer layer. The heatmap shows attention in all four quadrants, representing the bidirectional interactions between past context $[\mathcal{C}]$ and future query $[\mathcal{Q}]$ tokens. Brighter colors, indicating stronger attention, are visible for queries attending to the past context (bottom-left quadrant) but also for queries attending to other queries (bottom-right). This explicitly shows the model is learning the complex, bidirectional relationships and not just the standard query-to-context flow, enabled by our unified attention mechanism.

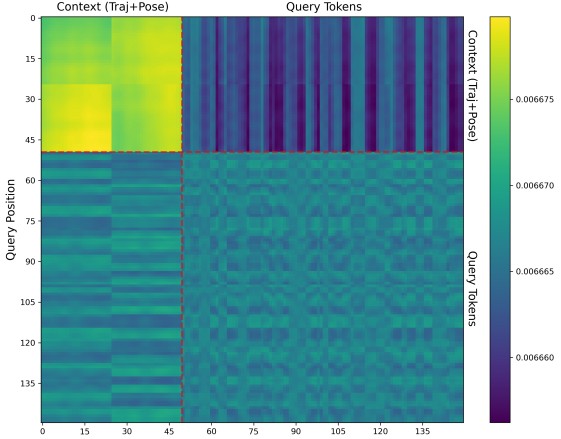

Figure 5: Attention patterns in the first transformer block at epoch 100. Brighter colors indicate stronger attention weights. Dashed lines (- - -) separate past context from future query tokens.

This visual evidence supports our hypothesis that this richer, bidirectional attention is key to our model's effectiveness.

### G.3 ARCHITECTURAL COMPONENT ABLATION

We investigate the contribution of our smaller architectural choices. Table 12 analyzes the impact of removing the Type Embedding ($\epsilon$) and replacing RMSNorm with LayerNorm on the MOCAP-

Table 12: Extended ablation study of architecture choices on MOCAP-UMPM. We explicitly compare RMSNorm vs. LayerNorm and the effect of Type Embeddings. Lower values are better ($\downarrow$), with the best results shown in **bold**.

| Components | | | Ours (wide) | | Ours (deep) | |
|---|---|---|---|---|---|---|
| **RMSNorm** | **LayerNorm** | **Type Emb.** | **APE$\downarrow$** | **JPE$\downarrow$** | **APE$\downarrow$** | **JPE$\downarrow$** |
| | ✓ | ✓ | 126.24 | 213.85 | 132.09 | 212.13 |
| ✓ | | | 127.37 | 213.06 | 132.80 | 214.85 |
| ✓ | | ✓ | **125.70** | **212.72** | **131.41** | **211.76** |

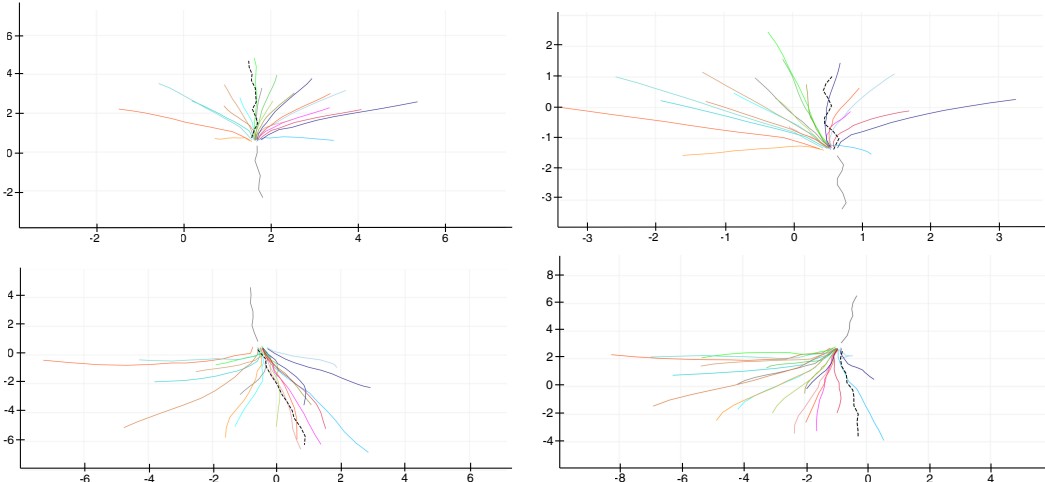

Figure 6: Visualization of trajectory predictions ($K = 20$) of our (wide) model on ETH-UCY data. X-Y coordinate values in the plots are in meters. All model predictions are in color, ground truth future trajectories are black dashes, and input trajectories are in gray.

UMPM dataset. The results confirm their importance: removing the type embeddings degrades performance (*e.g.*, APE increases from 125.70 to 126.24 for the wide model), confirming they are valuable for helping the model distinguish between pose and trajectory streams. RMSNorm also provides a consistent, albeit minor, performance benefit over LayerNorm while being more computationally efficient.

# H ADDITIONAL QUALITATIVE RESULTS

## H.1 TRAJECTORY-ONLY VISUALIZATION

To provide qualitative results for the trajectory-only task, Figure 6 visualizes our model's $K = 20$ proposals on challenging, crowded scenes from the ETH-UCY dataset. The visualizations show our model's ability to capture a wide, multi-modal distribution of plausible future paths, correctly identifying diverse outcomes (*e.g.*, turning left, turning right, or stopping) in high-uncertainty scenarios.

## H.2 FAILURE CASE ANALYSIS

While our model is robust, it is not without limitations, particularly in complex multi-person scenes. Our model treats all individuals independently, which can lead to unrealistic predictions when agents' motions are strongly coupled or highly unusual. Figure 7 presents qualitative failure cases from MOCAP-UMPM. The first example showcases an intricate interaction where the blue and green agents are turning in a circle while holding hands. Our model, processing them independently, fails to capture this complex, coupled motion. This also highlights that integrating explicit multi-

| $t = 0.4s$ | $t = 0.8s$ | $t = 1.2s$ | $t = 1.6s$ | $t = 2.0s$ |
| --- | --- | --- | --- | --- |

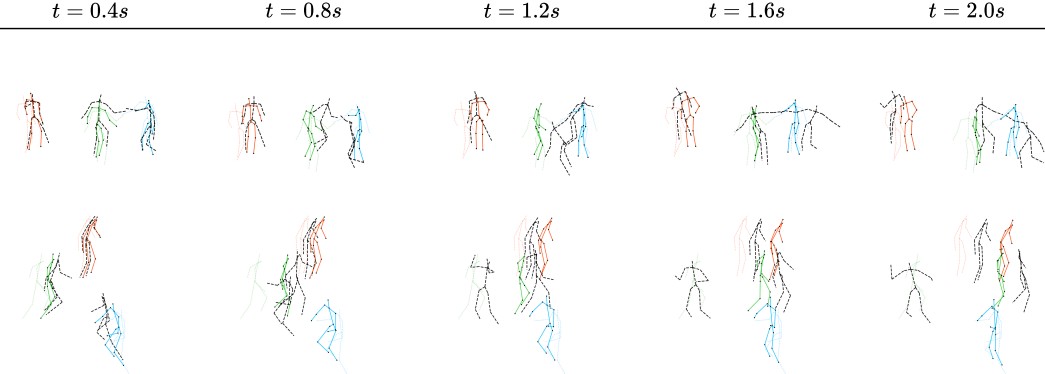

Figure 7: Visualization of predictions of our (wide) model on MOCAP-UMPM data. All model predictions are in color. Ground truth future poses are black dashes, and the last-known input positions are colored dashes.

agent interaction modules is a critical and promising direction for future work. The second example shows the blue and green agents undergoing an unexpected, rapid acceleration. The model's predictions struggle to keep pace with this abrupt change in dynamics, likely defaulting to a smoother or more mean-reverting trajectory, and thus accumulating significant error.

# I PERFORMANCE ON HIGHLY INTERACTIVE SCENARIOS

To specifically address the model's generalization capability on complex multi-agent interactions, we conducted an evaluation on the challenging WorldPose dataset (Jiang et al., 2024). Unlike the standard pedestrian dynamics in ETH-UCY or social mingling in UMPM, WorldPose features high-intensity sports scenarios (soccer) characterized by rapid changes in velocity, complex contact interactions, and adversarial intent. This serves as a rigorous stress test for our architecture in scenarios where prior work typically relies on dedicated interaction modules.

## I.1 EXPERIMENTAL SETUP

The WorldPose data was preprocessed to use an 80% train and 20% test split. The dataset contains player poses ($M = 24$) recorded at 25 fps at soccer games. All models observed 1.0 seconds of past motion and predicted the best of $K = 10$ proposals for 1.0 seconds into the future.

## I.2 RESULTS AND ANALYSIS

The results in Table 13 demonstrate that our simple, unified framework substantially outperforms prior complex, interaction-aware methods. Specifically, SimpliHuMoN reduces APE by 56.7% compared to T2P and 64.6% compared to EMPMP, likely due to the difficulty of modeling rapid, non-cyclic sports motions using architectures optimized for smoother walking gaits. Our method's ability to handle this data without architectural modification highlights the universality of the proposed Transformer decoder, validating the strength of our core architecture while underscoring that integrating explicit multi-agent interaction mechanisms remains a critical and promising direction for future work.

Table 13: Comparison of APE/JPE metrics on WorldPose data. Lower values are better (↓), with the best results shown in **bold**. An asterisk (*) denotes models we recomputed for this setup.

| Model | APE ↓ | JPE ↓ |
| --- | --- | --- |
| T2P* | 362.7 | 913.6 |
| EMPMP* | 443.4 | 981.5 |
| Ours (wide) | **156.8** | **746.3** |

