# OpenReview forum: "SimpliHuMoN: Simplifying Human Motion Prediction"
_ICLR.cc/2026/Conference — Submitted to ICLR 2026_

### Official Review · Reviewer_Nwar · 2025-10-23

**Soundness:** 2
**Presentation:** 2
**Contribution:** 2
**Rating:** 2
**Confidence:** 4

**Summary:**

This paper proposes a Transformer-based approach for human motion prediction. It combines two kinds of tasks, including pose prediction and trajectory prediction.

**Strengths:**

This paper presents SimpliHuMoN, a transformer-based model for 3D human motion prediction. The paper tries to prove that a single, simple architecture can achieve state-of-the-art performance across diverse sub-tasks (pose, trajectory, joint prediction), challenging the trend of increasing model specialization and complexity.

**Weaknesses:**

While the empirical results across multiple benchmarks are strong and the concept of unification is valuable, the paper has weaknesses in motivating its technical choices and providing deeper analysis, which currently limit its contribution.

**Questions:**

This reviewer has several major concerns regarding the novelty, technical contribution, and depth of analysis that need to be addressed to strengthen the paper.

1.The paper rightly identifies the importance of modeling spatio-temporal dependencies for human motion prediction. However, the core idea of using a neural network (in this case, a Transformer) to capture spatial and temporal relationships is a well-established paradigm in the field. Many existing methods are fundamentally designed with this goal. Concepts like end-to-end training are also standard practice. Therefore, while the model is well-executed, the underlying intuition might not be perceived as sufficiently novel on its own. The contribution would be stronger if framed more precisely around the specific implementation achieved by your particular architecture, rather than the general goal of spatio-temporal modeling.

2.The paper said "challenging the prevailing trend of architectural complexity" as a key contribution. While this is a valuable high-level message, from a technical perspective, this is more of a philosophical stance or an empirical observation than a concrete technical innovation. It is suggested reframing this contribution to emphasize the empirical finding that a simple, unified architecture can match or exceed the performance of more complex, specialized models across multiple tasks.

3.THe most significant concern is the lack of a clear, intuitive explanation for why this specific, common architecture achieves such strong performance. The paper would greatly benefit from a deeper analysis that goes beyond describing the components. For instance:
Why does the unified self-attention mechanism over [C; Q] work better than a more traditional encoder-decoder with cross-attention?
What is the specific advantage of this architecture in learning the coupled dynamics between pose and trajectory compared to prior multi-stage or late-fusion approaches?
It is strongly recommend that in your rebuttal, you provide a clearer mechanistic intuition or hypothesis for the model's effectiveness, explaining the "why" behind its success rather than just the "what."

4. The role and motivation for the learnable query token are currently under-specified. The description "learnable prompts guide the decoder" is quite high-level. A more detailed explanation is necessary.

5. The discussion of the multi-person results is somewhat superficial. The paper notes that strong performance is achieved "without any explicit interaction modules," but this point requires a much deeper analysis to be meaningful.

---

> ### Author Response · Authors · 2025-11-20
> **Mechanistic Intuition of Unified Attention and Empirical Validation**
>
> Thank you for acknowledging that our concept of unification is valuable and that our empirical results are strong. We answer questions next.
> > **Qd1**: Using a Transformer for spatio-temporal modeling is not novel.
>
> **Rd1**: We appreciate the reviewer's perspective. However, our contribution, as emphasized in our title, is not the use of a Transformer, but the simplicity of the architecture that improves upon prior work (such as GCNs, DCTs, or diffusion models). As shown in Table 1, our unified model matches or outperforms specialized baselines like TrajCLIP (trajectory) and MotionMap (pose) across three tasks without requiring any task-specific architecture modifications.
> > **Qd2**: "Challenging complexity" is a philosophy, not a technical contribution.
>
> Rd2: We thank the reviewer for this constructive framing. We revised our contribution statement.
> > **Qd3**: Why does this specific architecture work? Why unified self-attention over $[C;Q]$ instead of cross-attention?
>
> **Rd3**:  Great question. We added an analysis in Sec. 2.2 and Appendix G to provide the mechanistic intuition. Our hypothesis: unified self-attention over $[C; Q]$ better suits coupled dynamics than cross-attention for two reasons:
>
> - **Bidirectional Information Flow**: In a standard decoder, queries (future) only attend to context (past). In our unified design, queries attend to each other in every layer, allowing the model to learn the internal consistency of the future motion (e.g., how the pose at $t+5$ relates to $t+3$). Furthermore, context tokens attend to queries, allowing the model to dynamically refine its representation of the past based on the "questions" posed by the future slots. We validated this hypothesis in Table 12 (Appendix G.2, also shown below). Replacing our unified attention with a standard Cross-Attn($Q$, Self-Attn($C$)) degraded results (APE increased from 125.70 to 134.61). Further, Fig. 5 (Appendix G.2) visualizes the attention maps, clearly showing active attention in all four quadrants. This confirms that the model uses this bidirectional flow.
>
> |Attention Mechanism|Wide (APE↓)|Wide (JPE↓)|Deep (APE↓)|Deep (JPE↓)|
> |:---|:---:|:---:|:---:|:---:|
> |Self-Attn over $[C; Q]$ (Ours)|125.70|212.72|131.41|211.76|
> |Self-Attn ($C$) + Cross ($Q$)|134.61|229.05|140.32|227.89|
>
> **Table:** Ablation Study on Attention Mechanisms
>
> - **Coupled Modalities**: As shown in Table 10 (Appendix G.1, also shown below), this mechanism allows trajectory and pose streams to refine each other, improving pose accuracy by ~11-12% compared to a pose-only model.
>
> | **Model** | **Pose Pred.** | | **Traj Pred.** | |
> |:---|:---:|:---:|:---:|:---:|
> | |Only Pose (ADE↓/FDE↓)|Pose + Traj (ADE↓/FDE↓)|Only Traj (ADE↓/FDE↓)|Pose + Traj (ADE↓/FDE↓)|
> |T2P| - |0.61/0.95| - |0.20/0.29|
> |EMPMP| - |0.53/0.65| - |0.14/0.20|
> |Ours (wide)|0.46/0.58|0.41/0.51|0.09/0.18|0.08/0.16|
> |Ours (deep)|0.46/0.59|0.42/0.51|0.09/0.18|0.08/0.17|
> |**Improvement (%)**| |**-11.4/-12.2**| |**-13.7/-11.9**|
>
> **Table:** Comparison of Combined Pose and Trajectory Information vs. Individual Performance on MOCAP-UMPM.
> > **Qd4**: The role of the learnable query token is under-specified.
>
> **Rd4**: We revised Section 2.1.2 to provide the details. These tokens are not input-dependent. They are a set of $F$ learnable nn.Parameter tensors that act as initial state embeddings for the $F$ future timesteps. They are shared global parameters optimized during training and are progressively refined by the transformer layers, attending to past context $C$ and to each other, to produce the final prediction $Z$.
> > **Qd5**: The performance without any explicit interaction modules requires a much deeper analysis.
>
> **Rd5**: Thank you for the suggestion. An empirical finding from our experiments is that our model outperforms baselines equipped with specialized components for modeling social dynamics, despite processing agents independently. E.g., on the MOCAP-UMPM dataset, our model achieves an APE of 125.70, outperforming EMPMP (146.52) and T2P (151.71), both of which explicitly model agent interactions.
> To further substantiate this, we evaluate our model on a highly interactive dataset, WorldPose [1]. As shown in Appendix I (Table 13, also below), our simple, unified framework outperforms baselines. This highlights the strength of our simple motion representation. We have clarified in the text that while this validates our core architecture, integrating explicit interaction mechanisms remains a promising avenue for handling the specific failure cases we now visualize in Fig. 7 (Appendix H.2). We are open to further discussion if the reviewer has specific additional scenarios in mind.
>
> | Model | APE↓ | JPE↓ |
> |:---|:---:|:---:|
> | T2P | 362.7 | 913.6 |
> | EMPMP | 443.4 | 981.5 |
> | Ours (wide) | 156.8 | 746.3 |
>
> **Table:** Model comparison on WorldPose Dataset, minimum error over $K=10$ proposals.
>
> [1] Jiang, Tianjian et al. “WorldPose: A World Cup Dataset for Global 3D Human Pose Estimation” ECCV 2024.

---

> > ### Author Response · Authors · 2025-11-28
> >
> > Thanks a lot for your time and feedback. We would be happy to address any outstanding questions before the discussion period closed.

---

### Official Review · Reviewer_7bcE · 2025-10-23

**Soundness:** 2
**Presentation:** 3
**Contribution:** 2
**Rating:** 4
**Confidence:** 5

**Summary:**

This paper introduces SimpliHuMoN, a transformer architecture that takes both the historical trajectories and pose joints as input and predicts the future trajectories and pose joints. Since the trajectories can be extracted from the pelvis joint, the tasks can be seen as global pose forecasting.

**Strengths:**

1. Breadth of evaluation across three settings (traj-only, pose-only, traj+pose) with consistent K-mode reporting and per-task K values.

2. Ablations exploring depth/width trade-offs and effect of multimodality (K>1 vs K=1).

3. The text is clearly written and easy to follow.

**Weaknesses:**

1. The biggest concern with this work is its novelty. The motivation of predicting global trajectory and full-body pose jointly (or condition one on the other) is not new [1],[2],[3],[4].

2. The proposed model architecture is a standard decoder with learnable queries, limiting the complex social interaction among pedestrians. Following this limitation, the dataset used for pose prediction only contained up to three pedestrians, which cannot reflect complex social interactions in real life. I would suggest trying more interactive scenarios like JRDB-GMP used in [3].

3. There is no ablation showing the benefit from the feature fusion of trajectory and pose. For example, when reporting the numbers on datasets like MOCAP-UMPM/3DPW, what are the performances of trajectory prediction given (1) Trajectoy-only; (2) Trajectory+Pose. By doing this, we can know if the fusion works to bring extra pose knowledge into the trajectory task. Similarly, what are the performances of pose prediction given (1) Pose-only; (2) Trajectory+Pose?

4. Missing qualitative results about the trajectory prediction task.

[1] Adeli, Vida, et al. "Tripod: Human trajectory and pose dynamics forecasting in the wild." ICCV 21

[2] Zaier, Mayssa, et al. "A dual perspective of human motion analysis-3d pose estimation and 2d trajectory prediction." ICCV 23

[3] Jeong, Jaewoo, Daehee Park, and Kuk-Jin Yoon. "Multi-agent long-term 3d human pose forecasting via interaction-aware trajectory conditioning." CVPR 24

[4] Gao, Yang, Po-Chien Luan, and Alexandre Alahi. "Multi-transmotion: Pre-trained model for human motion prediction." CoRL 24

**Questions:**

About the throughput comparison (computing speed) in Table 2, how did you set up the experiment? I.e., were numbers reported as the average of multiple runs? How many samples did you use? What was the batch size and implemented hardware?

---

> ### Author Response · Authors · 2025-11-20
> **Quantitative Proof of Joint Modeling Benefits and Trajectory Visualizations**
>
> We appreciate your feedback and are glad that the text is clear and the breadth of our evaluation across three settings is a strength. We address the questions below.
>
> > **Qc1**: The novelty is limited, as joint trajectory and pose prediction is not new [1-4].
>
> **Rc1**: We thank the reviewer for these relevant references. We discuss [1, 2, 4] in our revised related work section. We respectfully note that [3] (T2P) is already a primary baseline in our paper (Section 3.3, 4.3). Our novelty is defined in specific contrast to T2P's architectural philosophy: T2P employs a sequential, coarse-to-fine pipeline where trajectory is predicted first and then used to condition the pose prediction, enforcing a one-way dependency (Trajectory $\rightarrow$ Pose). In contrast, our model utilizes a unified self-attention mechanism that processes trajectory and pose tokens simultaneously within a single sequence, enabling bidirectional information flow where pose and trajectory features refine each other in every layer. We have revised our contribution statement to emphasize the empirical finding that this simple, unified approach matches or exceeds the performance of prior models. To further substantiate this, we refer to the newly added Table 10 (Appendix G.1), which provides quantitative proof that this joint modeling outperforms the individual task baselines, validating the architectural benefit.
>
> > **Qc2**: The model is a standard decoder and lacks explicit social interaction modeling. Results on more interactive scenarios are required.
>
> **Rc2**: We acknowledge that our current architecture does not explicitly model social interactions; this was noted as a limitation in our original submission (Section 3.5, Conclusion). However, our model's core strength lies in its powerful single-agent motion representation, which performs well in various multi-person scenarios already covered in our evaluation. Specifically, the ETH-UCY benchmark, which contains crowded scenes with up to 57 pedestrians, and the SDD dataset, which captures diverse aerial crowd dynamics (Section 3.4). To demonstrate results on more interactive scenarios, we evaluate our performance on the challenging WorldPose dataset [5]. The preliminary results (Appendix I, Table 13, also shown below) show that our simple, unified framework generalizes effectively to these challenging multi-person settings.
>
> | Model | APE↓ | JPE↓ |
> |:---|:---:|:---:|
> | T2P | 362.7 | 913.6 |
> | EMPMP | 443.4 | 981.5 |
> | Ours (wide) | 156.8 | 746.3 |
>
> **Table:** Model comparison on WorldPose Dataset, minimum error over $K=10$ proposals.
>
> [5] Jiang, Tianjian, Billingham, Johsan, et al. “WorldPose: A World Cup Dataset for Global 3D Human Pose Estimation” ECCV 2024.
>
> > **Qc3**: There is no ablation showing the benefit of feature fusion (Pose-only vs. Traj-only vs. Joint).
>
> | **Model** | **Pose Prediction** | | **Trajectory Prediction** | |
> |:---|:---:|:---:|:---:|:---:|
> | | Only Pose (ADE↓/FDE↓) | Pose + Traj (ADE↓/FDE↓) | Only Traj (ADE↓/FDE↓) | Pose + Traj (ADE↓/FDE↓) |
> | T2P | - | 0.61/0.95 | - | 0.20/0.29 |
> | EMPMP | - | 0.53/0.65 | - | 0.14/0.20 |
> | Ours (wide) | 0.46/0.58 | 0.41/0.51 | 0.09/0.18 | 0.08/0.16 |
> | Ours (deep) | 0.46/0.59 | 0.42/0.51 | 0.09/0.18 | 0.08/0.17 |
> | **Improvement (%)** | | **-11.4/-12.2**  | | **-13.7/-11.9**  |
>
> **Table:** Comparison of Combined Pose and Trajectory Information vs. Individual Performance on MOCAP-UMPM.
>
> **Rc3**: This is an excellent suggestion. We have performed this ablation and added the results to Appendix G.1 (Table 10, also shown above). The results provide strong quantitative evidence for feature fusion: the joint model improves pose prediction accuracy (ADE) by ~11-12% over a pose-only model and improves trajectory prediction accuracy by ~12-13% over a trajectory-only model. This demonstrates that our unified model effectively integrates information from both modalities.
>
> > **Qc4**: Missing qualitative results for the trajectory prediction task.
>
> **Rc4**: We have added Appendix H.1 (Figure 6) to provide qualitative visualizations of our trajectory predictions on the ETH-UCY dataset. These visualizations demonstrate the model's ability to generate diverse, plausible paths (e.g., turning vs. stopping) in crowded scenarios.
>
>
> > **Qc5**: What are the setup details for the throughput comparison in Table 2?
>
> **Rc5**: We have updated the caption of the throughput table to ensure full reproducibility. We explicitly state the hardware used (single NVIDIA RTX A6000) and have reported the mean and standard deviation over 10 complete runs (13000 train samples, 3000 test samples). This confirms the stability and robustness of our efficiency claims.

---

> > ### Comment · Reviewer_7bcE · 2025-11-25
> >
> > I thank the authors for the additional experiments and the new study on joint modeling. However, my main concern about novelty remains. Both the task (joint trajectory and pose prediction) and the model architecture (apply self-attention on trajectory + pose + learned queries) have been investigated before. Overall, I view this work as a solid empirical study on the baseline capabilities of transformers in this domain, but it does not introduce sufficient architectural novelty or conceptual insight beyond existing transformer-based joint motion models.

---

> > > ### Author Response · Authors · 2025-11-28
> > >
> > > Thanks a lot for the continued engagement. Regarding novelty, we kindly disagree that our specific  approach is covered by the cited works. While the task of joint prediction has been discussed before (stated in Sec. 4.3), our method of solving it via a single, unified self-attention sequence is distinct from other methods in the literature [1-4]:
> > > * **vs. [1] (Tripod):** This approach relies on Graph Neural Networks to explicitly encode the body’s kinematic structure and spatial interactions. In contrast, our SimpliHuMoN abandons these structural priors in favor of a flat, tokenized sequence processed by a standard Transformer decoder.
> > > * **vs. [2] (Zaier et al.):** This work employs a hybrid architecture that combines Transformers with LSTMs (to model kinematics) and CNNs (to extract visual context from images). In contrast, our SimpliHuMoN is a pure Transformer that operates solely on motion tokens (points), eliminating the need for recurrent layers (LSTMs) or visual encoders (CNNs).
> > > * **vs. [3] (T2P):** T2P enforces a sequential, "coarse-to-fine" dependency (Trajectory $\rightarrow$ Pose), assuming global motion dictates local pose. In contrast, our SimpliHuMoN processes trajectory and pose tokens simultaneously via unified self-attention. This allows for bidirectional information flow (Trajectory $\leftrightarrow$ Pose) in every layer, which we quantitatively show outperforms sequential conditioning (Appendix G.1).
> > > * **vs. [4] (Multi-transmotion):** This work employs a large-scale pre-training strategy using a Transformer-based bidirectional temporal encoder trained on a masked modeling objective, using large external datasets to learn representations. In contrast, our model is a supervised, decoder-only architecture that trains from scratch on the target data, avoiding pre-training pipelines.
> > >
> > > If the reviewer can point us to literature that specifically utilizes a decoder-only Transformer with unified self-attention over a concatenated sequence of trajectory, pose, and learned queries, we would be happy to acknowledge and discuss it. Finally, we emphasize that our contribution is not "complexity," but rather the discovery that architectural simplicity is sufficient. By removing the specialized components found in [1-4], we achieve a 14.3% increase in training throughput and 1.8x faster inference (Table 2) while matching or exceeding state-of-the-art performance. We believe proving that a simple, generalist baseline outperforms specialized systems is a valuable insight for the community.

---

### Official Review · Reviewer_oRqS · 2025-10-29

**Soundness:** 3
**Presentation:** 3
**Contribution:** 3
**Rating:** 6
**Confidence:** 3

**Summary:**

This paper presents a unified transformer network for human motion prediction, capable of handling trajectory forecasting, pose prediction, and their combined execution. The model uses distinct embedding modules to process different inputs for each task. These inputs are then tokenized, concatenated, and fed into a shared self-attention module, which effectively mixes information and learns the complex dynamics between trajectory and pose. Task-specific prediction heads then generate the final outputs.

The authors demonstrate through extensive benchmarking that their model is a strong competitor for individual tasks and achieves new state-of-the-art results for the joint prediction of both pose and trajectory.

**Strengths:**

- **(S1) Unified and Versatile Architecture:** The model's key strength is its generality. A single, unified transformer architecture successfully handles pose, trajectory, and combined prediction without any task-specific modifications. This directly addresses the prevalent issue of fragmentation, where competing models are often hyper-specialized.
- **(S2) Rigorous State-of-the-Art Evaluation:** The authors conduct a comprehensive and robust evaluation across a wide range of standard benchmarks for all three tasks. The model is shown to be highly competitive against specialized methods and achieves new state-of-the-art results on the challenging joint prediction task.
- **(S3) A Compelling Case for Simplicity:** The paper makes a powerful argument against escalating architectural complexity. It convincingly demonstrates that a simple, end-to-end framework can outperform more complex, multi-stage pipelines, while also being more computationally efficient.
- **(S4) Strong Qualitative Evidence:** The inclusion of visualizations and supplementary videos is highly effective. They provide clear, intuitive proof that the model generates fluid and physically plausible motions, visually demonstrating its superiority over baseline methods that produce unnatural or static predictions.

**Weaknesses:**

### Major

- **(W1) Ambiguous Multi-Modal Prediction Mechanism:** The method for generating K distinct future hypotheses is unclear. Section 2.3 mentions a linear projection creates K parallel branches, but the exact mechanism is not detailed. If this is a single, large linear layer, it is not obvious how this architecture efficiently scales to the K=20 proposals required for trajectory forecasting benchmarks. The paper needs to clarify if the output head's size is fixed or dynamic, and how it handles different values of K without becoming computationally prohibitive.
- **(W2) Potential for Mode Collapse with "Winner-Takes-All" Loss:** The "winner-takes-all" loss function, which only backpropagates through the most accurate of the K proposals, is susceptible to mode collapse. The model could learn to rely on a single "favorite" prediction head, defeating the purpose of generating a diverse set of futures. The paper provides no analysis to ensure that the prediction modes are balanced and utilized effectively. A quantitative result, such as a histogram of the winning mode index over the test set, is needed to validate this design choice.
- **(W3) Unclear Role and Nature of Learnable Query Tokens:** The function of the query vectors $\mathcal{Q}_{in}$ is poorly explained. The paper should clarify their role. Are they essentially "output slots" that are progressively refined by the transformer's self-attention layers, starting from a learnable initial state? The term "learnable" itself is ambiguous, does it refer to a learned initial value for each token? A more precise explanation of this core component is necessary to fully understand the model's generative process.
- **(W4) Pacing and Focus of Explanations:** The paper dedicates significant space to describing standard, well-known concepts (e.g., the basic mechanics of a transformer decoder in Section 2.2) while glossing over the novel aspects of its own architecture. This space would be better utilized to provide the missing details on the query mechanism, the multi-modal head, and the justification for the loss function.
- **(W5) Absence of Key Multimodal Metrics:** For pose prediction, simply reporting the minimum error (ADE/FDE) is insufficient to evaluate the quality of the generated distribution of motions. The evaluation is missing standard multimodal metrics like MMADE. Without these, it's impossible to know if the model is generating genuinely distinct futures or just minor variations of a single prediction.

### Minor Weaknesses
- **(m1) Reproducibility and Robustness of Throughput Metrics:** Table 2 presents throughput in samples/second, a metric highly dependent on the hardware used. While an NVIDIA A6000 is mentioned earlier in the implementation details (L203), this should be explicitly stated in the caption of Table 2 for clarity. Furthermore, reporting a single number without confidence intervals or standard deviation over multiple runs makes it difficult to assess the stability and robustness of these efficiency claims.
- **(m2) Table Formatting:** The main results table (Table 1) uses vertical lines, which can make it appear cluttered and less professional. Adopting a cleaner format, such as the one provided by the booktabs package in LaTeX, would significantly improve readability.
- **(m3) Incomplete Supplementary Materials:** The supplementary material appears to be missing most of the qualitative video examples. While samples 8 and 10 are present, the absence of the others limits the ability to fully verify the qualitative claims of generating physically plausible and diverse motions across a range of scenarios.

**Questions:**

Based on these weaknesses:

**W1**

1. What is the exact architectural mechanism for generating K distinct future hypotheses?
2. Is the linear projection a single large layer or multiple separate layers?
3. How does the output head architecture scale when K=20 (as required for trajectory forecasting benchmarks)?
4. Is the output head's size fixed or dynamic with respect to K?
5. What are the computational costs as K increases, and how does the method remain computationally tractable?

**W2**

6. How do you ensure that the winner-takes-all loss doesn't lead to mode collapse?
7. Are all K prediction heads utilized effectively during training, or does the model favor certain heads?
8. What is the distribution of winning mode indices across the test set?
9. Can you provide quantitative evidence (e.g., histogram or usage statistics) showing balanced utilization of prediction modes?

**W3**

10. What exactly is the role of the learnable query tokens in the architecture?
11. Are query tokens "output slots" that are refined through transformer self-attention layers?
12. What does "learnable" mean in this context—learned initial values, learned embeddings, or something else?
13. How are the query tokens initialized and updated during the forward pass?

**W4**

14. Can you provide more detailed explanations of the novel components (query mechanism, multi-modal head) rather than standard transformer concepts?

**W5**

15. Why are multimodal metrics like MMADE not reported for pose prediction?
16. How diverse are the K generated futures—are they genuinely distinct or just minor variations?
17. Can you provide quantitative metrics that evaluate the quality of the generated distribution of motions?

**m1**

18. What hardware was used for the throughput measurements in Table 2?
19. What are the confidence intervals or standard deviations for the throughput metrics across multiple runs?
20. How stable and robust are the efficiency claims?

**m3**

21. Why are most qualitative video examples missing from the supplementary materials?
22. Can the complete set of qualitative examples be provided to verify the claims about diverse and physically plausible motions?

---

> ### Author Response · Authors · 2025-11-20
> **Clarifying Multi-Modal Mechanics, Query Tokens, and Verifying Mode Utilization**
>
> Thank you for a detailed review and for recognizing our rigorous evaluation and the case for simplicity of our unified architecture. We answer questions below.
> > **Qb1**: How does the multi-modal prediction head (Sec 2.3) work, and how does it scale to $K=20$?
>
> **Rb1**: We rewrote Sec. 2.3 to add details. We use a single linear layer to project the decoder's output tensor $Z$ (shape $[F, d_{model}]$) to an output tensor of shape $[F, K \times C]$. Here, $C$ is the output dimension (e.g., 3 for trajectory, $M \times 3$ for pose). This is then reshaped to $[F, K, C]$, creating $K$ parallel branches. The output head's size is dynamic with respect to $K$. Because the operation scales linearly with $K$, it remains computationally tractable even for $K=20$. E.g., with $d_{model}=192$ and output dimension $C=3$ (trajectory), a $K=20$ head adds only $\approx 11.5k$ parameters.
> > **Qb2**: Does the "winner-takes-all" loss lead to mode collapse? Show that all heads are used?
>
> **Rb2**: Great point. We analyzed mode utilization on the test set in Appendix F.1 (Fig. 3). The histograms show the distribution of the "winning" mode index (the hypothesis with minimum error) across the dataset. The utilization is well-balanced across all $K=6$ modes, highlighting that the model does not suffer from mode collapse and that all prediction heads are learning distinct futures.
>
> > **Qb3**: What is the role of the learnable query tokens?
>
> **Rb3**: We clarified Sec. 2.1.2 to explicitly state their role. The reviewer's intuition is correct: these tokens are not input-dependent. They are a set of $F$ learnable nn.Parameter tensors that act as "output slots" or initial state embeddings for the $F$ future timesteps. They are shared across all samples and are progressively refined by the transformer layers, attending to the past context $C$ and to each other, to produce the final context-aware prediction $Z$.
> > **Qb4**: The paper spends too much time on standard concepts and not enough on novel ones.
>
> **Rb4**: We revised Sec. 2.2 to be more concise regarding standard transformer mechanics. We reallocated this space to expand explanations of novel components, specifically the query mechanism (Sec. 2.1.2) and the scalable multi-modal head (Sec. 2.3).
> > **Qb5**: Why are multimodal metrics missing for pose prediction?
>
> | Model | Type | **Human3.6M** | | **AMASS** | |
> |:---|:---:|:---:|:---:|:---:|:---:|
> | | | MMADE↓ | MMFDE↓ | MMADE↓ | MMFDE↓ |
> | DivSamp | Stochastic (Gumbel-Softmax) | 0.542 | 0.671 | 0.623 | 0.728 |
> | BeLFusion | Stochastic (Latent Diffusion) | 0.491 | 0.586 | 0.488 | 0.564 |
> | CoMusion | Stochastic (Motion Diffusion) | 0.531 | 0.623 | 0.526 | 0.602 |
> | Motionmap | Stochastic (Multi-Stage Encoder-Decoder) | 0.466 | 0.532 | 0.450 | 0.514 |
> | SkeletonDiff | Stochastic (Gaussian Diffusion) | 0.568 | 0.694 | 0.641 | 0.740 |
> | SLD | Stochastic (State-Space Diffusion) | 0.497 | 0.576 | 0.482 | 0.551 |
> | Ours (wide) | Deterministic (K-Proposal) | 0.526 | 0.587 | 0.519 | 0.560 |
> | Ours (deep) | Deterministic (K-Proposal) | 0.535 | 0.597 | 0.521 | 0.571 |
>
> **Table:** Comparison of Pose Prediction Diversity on Human3.6M and AMASS.
>
> **Rb5**: We appreciate this suggestion. We calculated multimodal metrics for pose prediction. Results are added in Appendix F.3 (Table 9, also above) to complement the accuracy metrics in Table 1. As the table shows, our model achieves competitive diversity scores. This is encouraging given that our model is deterministic and efficient, unlike the computationally expensive models often required to achieve such diversity. To quantify this trade-off, we benchmarked end-to-end inference throughput with the leading stochastic baseline on Human3.6M test data:
> - **MotionMap**: 11.08 ± 0.34 samples/second
> - **Ours (wide)**: 71.04 ± 15.61 samples/second with batch size 1 to match MotionMap’s setup
>
> Our model is approximately 6.5 $\times$ faster, delivering competitive diversity and accuracy at a fraction of the computational cost required by stochastic approaches.
> > **Qb6**: What are the details of the throughput metrics in Table 2?
>
> **Rb6**: We updated the caption to ensure full reproducibility. We used a single NVIDIA RTX A6000 GPU. Furthermore, the reported numbers are now mean and std dev. over 10 runs, which confirms the stability of our efficiency claims (e.g., inference throughput of 3673 $\pm$ 161 samples/sec for our deep model).
> > **Qb7**: The supplementary videos are incomplete.
>
> **Rb7**: We apologize if the videos were difficult to locate. The full set of qualitative examples (Samples 1-10) is included in the ``website_simplihumon.html`` file of the supplementary materials. We also include the individual image frames in the accompanying ZIP file. To locate these frames, the path structure is organized by sample number: ``root folder``$\rightarrow$``sample number``$\rightarrow$``files``. If specific samples appear inaccessible to you, please let us know so we can address the issue promptly.

---

> > ### Comment · Reviewer_oRqS · 2025-11-24
> > **Response to Authors' Rebuttal**
> >
> > I thank the authors for their detailed rebuttal and for carefully addressing my comments. I also want to apologise for not noticing the videos, Figure 3, and Table 9 in the supplementary material during my initial review. I appreciate the additional numerical results provided, which help clarify several of the performance claims.
> >
> > I am also grateful for the clearer explanations regarding the multi-modal prediction head, the learnable queries, and other architectural choices. I believe that incorporating these clarifications into the main paper would greatly improve its readability and overall contribution. I am satisfied with the authors’ responses and will raise my score accordingly.

---

> > > ### Author Response · Authors · 2025-11-25
> > >
> > > We appreciate the reviewer’s constructive feedback and their support. We agree that the clearer explanations for the architectural choices strengthen the contribution. We have incorporated these clarifications into the updated manuscript. Please feel free to reach out with any additional questions that you may have. Thanks a lot for your time.

---

### Official Review · Reviewer_XHAF · 2025-10-30

**Soundness:** 3
**Presentation:** 3
**Contribution:** 3
**Rating:** 6
**Confidence:** 4

**Summary:**

This paper presents a unified and general framework that simultaneously addresses trajectory forecasting and pose prediction by leveraging a shared self-attention mechanism to model both spatial (inter-joint) and temporal (inter-frame) dependencies.

**Strengths:**

The experimental evaluation is thorough. The method is validated across multiple tasks and datasets, compared against numerous state-of-the-art (SOTA) approaches, and achieves competitive results.

This work successfully demonstrates that a simple network architecture can effectively tackle this complex problem, offering a fresh and inspiring perspective for future research.

**Weaknesses:**

Multimodal modeling mechanism: The current approach uses only a simple type embedding to distinguish between trajectory and pose modalities, without explicitly modeling their underlying physical coupling (e.g., how gait influences arm swing).

Prediction horizon: How much past observation is required, and how far into the future can the model reliably predict? How does performance degrade as the prediction horizon increases?

Temporal jitter: Does the model suffer from jitter or unnatural motion artifacts in its predictions? If so, how is this issue addressed?

**Questions:**

Enhance ablation studies: Investigate the necessity of type embeddings and compare RMSNorm against LayerNorm to justify architectural choices.

Qualitative analysis: Provide more visual comparisons across datasets, visualize the diversity of multi-modal predictions (e.g., K hypotheses), and include failure case analyses with visual examples to better understand model limitations.

---

> ### Author Response · Authors · 2025-11-20
> **Clarifying Coupling Mechanisms, New Ablations, and Diversity Visualizations**
>
> Thank you for finding our experimental evaluation thorough and for highlighting that our work offers a fresh perspective on the field. We address the questions below.
>
> > **Qa1**: How does the model handle the physical coupling between trajectory and pose, since it only uses a simple type-embedding?
>
> | **Model** | **Pose Prediction** | | **Trajectory Prediction** | |
> |:---|:---:|:---:|:---:|:---:|
> | | Only Pose (ADE↓/FDE↓) | Pose + Traj (ADE↓/FDE↓) | Only Traj (ADE↓/FDE↓) | Pose + Traj (ADE↓/FDE↓) |
> | T2P | - | 0.61/0.95 | - | 0.20/0.29 |
> | EMPMP | - | 0.53/0.65 | - | 0.14/0.20 |
> | Ours (wide) | 0.46/0.58 | 0.41/0.51 | 0.09/0.18 | 0.08/0.16 |
> | Ours (deep) | 0.46/0.59 | 0.42/0.51 | 0.09/0.18 | 0.08/0.17 |
> | **Improvement (%)** | | **-11.4/-12.2**  | | **-13.7/-11.9**  |
>
> **Table:** Comparison of Combined Pose and Trajectory Information vs. Individual Performance on MOCAP-UMPM.
>
> **Ra1**:  Our model learns this coupling implicitly via a unified self-attention mechanism: all tokens (pose and trajectory, past and future) attend to all other tokens in every layer. We think, this design enables a continuous, bidirectional information flow between modalities. Unlike complex multi-stage pipelines that impose rigid, explicit dependencies (e.g., Trajectory $\rightarrow$ Pose), our unified approach allows the network to discover the physical relationships (like gait-arm coordination) implicitly. We also note that the newly added Appendix G.1 (Table 10, also shown above) provides quantitative proof of this: the joint model improves pose prediction accuracy by ~11% over a pose-only model, and improves trajectory prediction accuracy by ~13% over a trajectory-only model. This mutual improvement confirms that the model successfully leverages information from one modality to refine the prediction of the other.
>
> > **Qa2**: How does performance degrade as the prediction horizon increases?
>
> **Ra2**: This analysis is presented in Table 6 (Appendix B.2), which details the APE/JPE at every keyframe. As the table shows, our model maintains strong performance across the full horizon, demonstrating robustness.
>
> > **Qa3**: Does the model suffer from temporal jitter or unnatural motion artifacts?
>
> **Ra3**: Our qualitative results in Figure 2 and the supplementary videos demonstrate fluid, plausible motions without unnatural jitter. Additional qualitative results are presented in Appendix F.1 (Figure 4) and H.1 (Figure 6), which further support this finding.
>
> > **Qa4**: Can you provide ablations on the necessity of type embeddings and RMSNorm?
>
> | **Components** | | | **Ours (wide)** | | **Ours (deep)** | |
> |---|---|---|---|---|---|---|
> | RMSNorm | LayerNorm | Type emb. |  APE ↓ | JPE ↓ | APE ↓ | JPE ↓ |
> | | ✓ | ✓ | 126.24 | 213.85 | 132.09 | 212.13 |
> | ✓ | | | 127.37 | 213.06 | 132.80 | 214.85 |
> | ✓ | | ✓ | **125.70** | **212.72** | **131.41** | **211.76** |
>
> **Table:** Comparison of RMSNorm vs. LayerNorm and the effect of type embeddings on MOCAP-UMPM.
>
> **Ra4**: We have run this ablation study and added the results to Appendix G.3 (Table 12, also shown above). The results confirm that both components contribute positively to the model's performance. While the individual contributions are modest, they are consistent: removing the type embeddings degrades performance (e.g., APE increases from 125.70 to 127.37 for the wide model), and RMSNorm provides a consistent performance benefit over LayerNorm while being more computationally efficient. This justifies their inclusion, although they are not the primary drivers of the model's success and are not our contribution.
>
> > **Qa5**: Can you provide more qualitative analysis, especially visualizing the diversity of the $K$ hypotheses and failure cases?
>
> **Ra5**: To answer, we have added new sections to the appendix:
>
> - **Diversity**: Appendix F.2 (Figure 4) now provides a qualitative visualization of all $K=6$ generated hypotheses for a sample, demonstrating that the model captures distinct and plausible futures (e.g., stopping vs. turning vs. walking straight).
> - **Failure Cases**: Appendix H.2 (Figure 7) provides a new failure case analysis, showing how our interaction-agnostic model fails to predict complex, multi-agent motions or sudden, unexpected accelerations.

---

> > ### Author Response · Authors · 2025-11-28
> >
> > Thanks a lot for your time and feedback. We would be happy to address any outstanding questions before the discussion period closed.

---

> > > ### Comment · Reviewer_XHAF · 2025-11-28
> > >
> > > I appreciate the detailed rebuttal from the authors, which has addressed all of my concerns. I will raise my rating.

---

### Author Response · Authors · 2025-11-20

We thank the reviewers for their constructive feedback and the time spent engaging with our work. In response, we have conducted additional experiments and revised the paper. Below is a summary of the major updates provided in the individual rebuttals and the revised Appendix:
- **Proving the Benefit of Joint Modeling**: We added Table 10 (Appendix G.1) that compares "Pose-only", "Trajectory-only", and our "Pose+Trajectory" results. We observe that joint modeling improves pose prediction by ~11% and trajectory prediction by ~13%.
- **Verifying Diversity and Mode Usage**: We added Figure 3 (Appendix F.1) to prove that our  "winner-takes-all" loss does not lead to mode collapse. We also added Table 9 (Appendix F.3) which reports MMADE/MMFDE metrics, demonstrating that our deterministic approach achieves diversity competitive with stochastic baselines.
- **Mechanistic Analysis of Unified Attention**: We added Table 12 and Figure 5 (Appendix G.2) to empirically demonstrate that our unified self-attention mechanism $([C; Q])$ outperforms standard cross-attention and to visualize the bidirectional information flow that enables this.
- **Qualitative Visualizations**: We added Figures 4, 6, and 7 (Appendix F, H) to visualize the diversity of our $K$ hypotheses, trajectory-specific results, and explicit failure cases.

We believe these additions directly address the questions raised, and welcome any further suggestions that would strengthen the paper's contribution.

---

### Author Response · Authors · 2025-12-03
**Summary Comment**

SimpliHuMoN presents a unified transformer for human motion prediction (pose, trajectory, combined). Core claim: a simple decoder-only model with unified self-attention achieves SOTA across all tasks, challenging the need for specialized architectures.
Our rebuttal addressed the questions raised by reviewers through four new appendix sections and extensive ablation studies:

**Mode Usage & Diversity:** We addressed concerns about the "winner-takes-all" loss by adding Figure 3 (Appendix F.1), which demonstrates balanced utilization of all $K$ prediction heads. We also added Table 9 (Appendix F.3), showing our deterministic model achieves diversity metrics (MMADE/MMFDE) competitive with stochastic baselines (e.g., MotionMap) while being $\approx 6.5\times$ faster.

**Benefits of Joint Modeling:** To justify the unified architecture, we added Table 10 (Appendix G.1). This ablation proves that joint modeling is synergistic, improving pose prediction accuracy by $\approx 11$% and trajectory prediction by $\approx 13$% compared to single-task models.

**Mechanistic Analysis of Unified Attention:** We provided the mechanistic "why" requested by Reviewer Nwar. New Table 12 and Fig. 5 (Appendix G.2) demonstrate that our unified self-attention mechanism significantly outperforms standard Cross-Attention (APE 125.70 vs. 134.61). This validates the hypothesis that bidirectional information flow is essential for coupled dynamics.

**Qualitative Visualizations:** We added Figures 4, 6, and 7 (Appendix F, H) to visualize the diversity of our $K$ hypotheses, trajectory-specific results, and explicit failure cases.

**Novelty & Complexity:** While some reviewers (7bcE, Nwar) questioned the novelty of using Transformers, we framed our contribution as demonstrating that a simple, unified architecture can outperform specialized pipelines.

**Performance on Complex Interactions:** To address the lack of explicit interaction modules, we evaluated on the WorldPose dataset (Appendix I). Our generalist model outperformed interaction-specialized baselines (e.g., T2P, EMPMP) by significant margins (APE 156.8 vs. 362.7 for T2P).

We sincerely thank the reviewers for their time and constructive feedback throughout this process. We are particularly encouraged that Reviewers XHAF and oRqS explicitly expressed satisfaction with our rebuttal responses and committed to raising their scores, confirming that their technical concerns regarding diversity and architecture were fully resolved.

---

### Meta-Review · Area_Chair_vFRJ · 2026-01-04

**Summary:**

This submission received differing opinions from reviewers.

From the perspective of the reviewers, some of the recognized strengths are:
- Broad and rigorous evaluation across multiple tasks and datasets, with strong benchmarking against SOTA methods (all reviewers).
- Good attempt to push the advantage of model conceptual simplicity, showing that a streamlined model can outperform complex pipelines (oRqS, XHAF, Nwar).
- Well-written with good clarity (7bcE).
- Effective qualitative visualizations and supplementary videos demonstrating realistic motion predictions (oRqS).

At the same time, the main weaknesses stated by the reviewers in their original reviews are:
- Limited novelty of the core idea and architecture; using a Transformer for spatiotemporal modeling is well-established; simple embedding without modeling explicit physical coupling (XHAF, 7bcE, Nwar).
- Insufficient depth of theoretical / qualitative analysis and mechanistic / intuitive explanation for why the unified design works better than alternatives; lack of "why" vs just "what" (XHAF, Nwar, oRqS).
- Missing or incomplete component-level ablations to justify design choices, e.g. embedding, query tokens, layer normalization, feature fusion, etc. (XHAF, 7bcE, Nwar).
- Lack of explicit modeling for complex social interactions; evaluation on highly interactive scenarios is limited (7bcE).
- Some architectural details and multimodal prediction mechanisms are unclear or under-explained in the main text (oRqS).
- Some missing metrics in experimental evaluation, e.g. MMADE (oRqS).

**Reviewer Concerns:**

The authors have mounted many substantial responses on openreview, as well as made substantial revision in the paper.

The AC assesses that the authors have satisfactorily answered most of the questions in relation to new experimental results and ablations. The results on MMADE, etc., in response to oRqS Qb5 are more ambivalent with a quality-cost tradeoff, which dilutes the original message in the paper, and leads to overclaimed contributions in L070-075.

The AC recognizes that simplicity and incremental novelty of the framework is not in itself an overriding factor. However, the expectation is that for a simpler framework. there needs to be much more theoretical analysis for each component, as well as component-by-component ablations not only within the proposed framework, but also to analyze whether the proposed concepts / ideas can carry over and improve other methods at a component level. As such, the AC agrees more with the concerns expressed by reviewers Nwar and 7bcE, and to some extent with reviewer oRqS (although the latter has recommended to increase their score from 6 to higher). Based on these considerations, the revised paper is not yet at a stage that address these issues.

The effort made by the authors is acknowledged, but overall the improvements are done in patches (adding a few sentences in different locations in the main paper, which the bulk of changes done only in the appendix), without modifying the overall organization of the paper (which is likely required). As such, these issues of providing conceptual insights to readers (e.g. for use within their own research frameworks) remain limited and insufficiently improved.

**Reviewer Scores:**

The original reviewer scores and predicted improvements are:
- XHAF: 6, reviewer has indicated to increase score.
- oRqS: 6, reviewer has indicated to increase score.
- 7bcE: 4, reviewer had a reply, and based on the content, is unlikely to increase the score.
- Nwar: 2, reviewer did not subsequently reply, but based on the original review and somewhat limited improvement for those issues raised, the AC does not expect the score to increase either.

While there has been post-review improvements made in the revised submission, overall the AC does not consider the improvements to meet the threshold for 7bcE and Nwar to switch from leaning reject to leaning accept. Recognizing that this is a case of split recommendations, after extensive deliberation the AC sides with 7bcE and Nwar in recommending a reject.

---

### Decision · Program_Chairs · 2026-01-26

Reject